# Heterozygosity for neurodevelopmental disorder-associated *TRIO* variants yields distinct deficits in behavior, neuronal development, and synaptic transmission in mice

Yevheniia Ishchenko[1,2]\*[†], Amanda T Jeng[1,3][†], Shufang Feng[1,4], Timothy Nottoli[5], Cindy Manriquez-Rodriguez[6], Khanh K Nguyen[6], Melissa G Carrizales[1], Matthew J Vitarelli[1], Ellen E Corcoran[1], Charles A Greer[2,3,7], Samuel A Myers[6], Anthony J Koleske[1,2,3]\*

[1]Department of Molecular Biophysics and Biochemistry, Yale University, New Haven, United States; [2]Department of Neuroscience, Yale School of Medicine, New Haven, United States; [3]Interdepartmental Neuroscience Program, Yale University, New Haven, United States; [4]Department of Gerontology, The Third Medical Center, Chinese PLA General Hospital, Beijing, China; [5]Department of Comparative Medicine, Yale School of Medicine, New Haven, United States; [6]Laboratory for Immunochemical Circuits, La Jolla Institute for Immunology, La Jolla, United States; [7]Department of Neurosurgery, Yale School of Medicine, New Haven, United States

\*For correspondence:
yevheniia.ishchenko@yale.edu (YI);
anthony.koleske@yale.edu (AJK)

[†]These authors contributed equally to this work

## eLife Assessment

This study explores how heterozygosity for specific neurodevelopmental disorder-associated TRIO variants affects brain function in mice. The authors conducted thorough analyses on mouse lines harboring TRIO-variants associated with autism spectrum disorder, schizophrenia, and bipolar disorder, and the results provide **compelling** evidence demonstrating unique alterations of each variant in synaptic functions and behavior. These findings highlight a **fundamental** aspect of TRIO variants contributing to brain functions and neuropsychiatric disorders.

**Abstract** Genetic variants in *TRIO* are associated with neurodevelopmental disorders (NDDs) including schizophrenia (SCZ), autism spectrum disorder (ASD), and intellectual disability. TRIO uses its two guanine nucleotide exchange factor (GEF) domains to activate GTPases (GEF1: Rac1 and RhoG; GEF2: RhoA) that control neuronal development and connectivity. It remains unclear how discrete *TRIO* variants differentially impact these neurodevelopmental events. Here, we investigate how heterozygosity for NDD-associated *Trio* variants – *+/K1431M* (ASD), *+/K1918X* (SCZ), and *+/M2145T* (bipolar disorder, BPD) – impacts mouse behavior, brain development, and synapse structure and function. Heterozygosity for different *Trio* variants impacts motor, social, and cognitive behaviors in distinct ways that model clinical phenotypes in humans. *Trio* variants differentially impact head and brain size, with corresponding changes in dendritic arbors of motor cortex layer 5 pyramidal neurons (M1 L5 PNs). Although neuronal structure was only modestly altered in the *Trio* variant heterozygotes, we observe significant changes in synaptic function and plasticity. We also identified distinct changes in glutamate synaptic release in *+/K1431M* and *+/M2145T* cortico-cortical synapses. The TRIO K1431M GEF1 domain has impaired ability to promote GTP exchange on Rac1,

but *+/K1431M* mice exhibit increased Rac1 activity, associated with increased levels of the Rac1 GEF Tiam1. Acute Rac1 inhibition with NSC23766 rescued glutamate release deficits in *+/K1431M* variant cortex. Our work reveals that discrete NDD-associated *Trio* variants yield overlapping but distinct phenotypes in mice, demonstrates an essential role for Trio in presynaptic glutamate release, and underscores the importance of studying the impact of variant heterozygosity in vivo.

## Introduction

Neurodevelopmental disorders (NDDs), including autism spectrum disorder (ASD), intellectual disability (ID), developmental delay (DD), schizophrenia (SCZ) and bipolar disorder (BPD), disrupt brain development and function. NDDs share considerable comorbidities and present with over-lapping symptoms, including impairments in cognition, behavior, language, social and motor skills, emotions, and learning ability (*American Psychiatric Association, 2013*). In addition, NDDs share many risk genes, further suggesting shared mechanistic underpinnings and pathology for these disorders (*Anttila et al., 2018*). A major challenge remains in understanding how diverse risk-conferring variants contribute to the pathophysiology of various NDDs.

*TRIO* encodes a large cytoskeletal regulatory protein with two guanine nucleotide exchange factor (GEF) domains for Rho family GTPases - GEF1 activates Rac1 and RhoG, while GEF2 activates RhoA (*Debant et al., 1996*; *Bellanger et al., 1998*; *Blangy et al., 2000*). TRIO relays signals from cell surface receptors, acting on GTPases to coordinate cytoskeletal rearrangements critical for proper neurodevelopment (*Grubisha et al., 2022*; *Paskus et al., 2020*; *Duman et al., 2015*; *Hall, 1998*; *Luo, 2000*; *Nakayama et al., 2000*; *Newey et al., 2005*; *Terry-Lorenzo et al., 2016*; *O'Neil et al., 2021*). *Trio* knockout mice exhibit decreased survival, skeletal muscle defects, and severe defects in brain development (*O'Brien et al., 2000*; *Peng et al., 2010*; *Zong et al., 2015*). Selective ablation of *Trio* in either excitatory or inhibitory neurons alters their morphology, impairs synaptic signaling, and yields NDD-related behavioral defects (*Katrancha et al., 2019*; *Sun et al., 2021*). Trio deficiency also leads to aberrations in long-term potentiation (LTP), as Trio-Rac1 signaling promotes AMPA receptor trafficking to increase synaptic strength (*Katrancha et al., 2019*; *Tian et al., 2021*; *Paskus et al., 2019*).

Damaging de novo mutations and ultra-rare variants in *TRIO* are enriched in individuals with NDDs (*Ba et al., 2016*; *Katrancha et al., 2017*; *Barbosa et al., 2020*; *Bonnet et al., 2023*; *Gazdagh et al., 2023*; *Pengelly et al., 2016*; *Sadybekov et al., 2017*; *Singh et al., 2022*). Interestingly, nonsense variants spread throughout *TRIO* are enriched in individuals with SCZ (*Singh et al., 2022*; *Howrigan et al., 2020*), whereas pathogenic missense *TRIO* variants in or surrounding the GEF1 domain are associated with ASD/ID (*Katrancha et al., 2017*; *Pengelly et al., 2016*; *Sadybekov et al., 2017*). Variants in the TRIO GEF1 domain that decrease Rac1 activity are associated with milder ID and microcephaly, whereas variants in the adjacent spectrin repeat 8 domain that increase Rac1 activity are associated with more severe ID and macrocephaly (*Barbosa et al., 2020*; *Bonnet et al., 2023*; *Kloth et al., 2021*; *Bircher et al., 2022*). Rare missense variants in *TRIO* have also been observed in BPD, epilepsy, and other disorders, but studies to date are underpowered to establish a causal link with these disorders. Given the wide spectrum of *TRIO* variants associated with different pathological conditions, a fundamental and unresolved question is the mechanisms by which distinct *TRIO* variants differentially impact normal mammalian brain development and function. NDD-associated *TRIO* variants affect in vitro GEF1 and GEF2 activities, cell morphology, inhibitory neuron maturation, axon guidance, and synapse function (*Katrancha et al., 2019*; *Sun et al., 2021*; *Tian et al., 2021*; *Bircher et al., 2022*; *Barbosa et al., 2020*; *Bonnet et al., 2023*); however, the impact of heterozygosity for distinct *TRIO* variants in vivo, as found in individuals with NDDs, has been unexplored.

We report here the comprehensive analysis of mice heterozygous for discrete *Trio* variants impacting Trio functional domains and associated with different NDDs: *+/K1431M* in the Trio GEF1 domain associated with ASD, *+/K1918X* leading to nonsense decay associated with SCZ, and *+/M2145T* in the Trio GEF2 domain, a de novo mutation found in an individual with BPD that impacts GEF2 activity. We show that these distinct *Trio* NDD-associated variants differentially impair mouse behavior, brain development, and inhibitory and excitatory synaptic transmission. We show, for the first time, that *Trio* variants that impact GEF1 and GEF2 activity differentially impact presynaptic neurotransmitter release and synaptic vesicle replenishment. In addition, we found that while the K1431M mutation impairs Trio GEF1 activation of Rac1 in vitro, *+/K1431M* mice exhibit increased levels of active Rac1. This was

associated with increased levels of the Rac1 GEF Tiam1, and acute Rac1 inhibition with NSC23766 rescued glutamate release deficits in *+/K1431M* variant cortex. Together, our data show how discrete heterozygous *TRIO* variants that differentially impact Trio biochemical activities yield divergent effects on behavior, neurodevelopment, and synaptic transmission.

## Results

### Generation of *Trio* variant mice and impact on *Trio* isoforms

To evaluate how different ways of impairing Trio could affect mammalian brain development and function, we used CRISPR/Cas9 technology to generate mice bearing heterozygous *Trio* variant alleles in different Trio functional domains (*Figure 1A and B*). We chose these alleles for their discrete and measurable effects on TRIO levels or GEF activity in vitro: K1431M impairs TRIO GEF1 interaction with and subsequent activation of Rac1 in vitro up to eightfold (*Figure 1—figure supplement 1A and B*; *Katrancha et al., 2017*; *Sadybekov et al., 2017*), M2145T TRIO GEF2 has a reduced ability to activate RhoA as a function of protein concentration in cells (*Katrancha et al., 2017*), and K1918X is predicted to lead to nonsense-mediated decay and loss of Trio protein (*Katrancha et al., 2017*). These *Trio* variant alleles are found heterozygous in individuals with NDDs: *+/K1431M* is associated with ASD, *+/K1918X* is associated with SCZ, and *+/M2145T* is a de novo mutation found in an individual with BPD (*Bircher and Koleske, 2021*).

Because *Trio* knockout mice exhibit decreased survival (*O'Brien et al., 2000*; *Peng et al., 2010*; *Zong et al., 2015*), we first verified the viability of mice bearing these CRISPR/Cas9-generated *Trio* alleles. Mice heterozygous for any one of these three *Trio* alleles survived to adulthood. Mice homozygous for *Trio K1431M* and *M2145T* survived to adulthood, with genotypes from offspring of *+/K1431M* intercrosses observed in Mendelian ratios, but fewer than expected *M2145T* homozygotes obtained from *+/M2145T* intercrosses (*Figure 1—figure supplement 1C*). *K1918X* homozygote pups were not observed (*Figure 1—figure supplement 1C*), as expected for a *Trio* null allele (*O'Brien et al., 2000*; *Zong et al., 2015*). We focused on heterozygotes, as most rare damaging *Trio* variants are heterozygous in humans.

Alternative splicing of the full-length *TRIO* transcript generates multiple brain-specific isoforms of various sizes, whose expression level varies by brain region and developmental age: cortex-predominant Trio9S (263 kDa) and Trio9L (277 kDa), brain-wide Duet (145 kDa), and cerebellum-specific Trio8 (217 kDa; *Figure 1A*; *McPherson et al., 2005*; *Portales-Casamar et al., 2006*). Heterozygosity for the *K1431M* and *M2145T* alleles did not alter the levels of the predominant Trio isoforms in the brain at postnatal day 0 (P0, neonate; *Figure 1C–D*) or at P42 (young adult; *Figure 1—figure supplement 1D–L*). Trio9 protein levels were reduced by ~50% in the brains of *+/K1918X* mice at P0 (*Figure 1C–D*) and P42 (*Figure 1—figure supplement 1D–L*), and we did not detect the presence of residual truncated protein (expected at 217 kDa), suggesting this *K1918X* mutation indeed leads to nonsense-mediated decay as predicted. Meanwhile, as the *K1918* site is not contained in these isoforms, the levels of Trio8 and Duet were unaffected (*Figure 1—figure supplement 1D–L*).

### *Trio* variant alleles differentially impact active Rho family GTPase levels

Having demonstrated that heterozygosity for the K1431M and M2145T alleles does not alter Trio protein levels in the brain, we can attribute any observed differences to the effects of these missense mutations on Trio GEF domain function rather than changes in Trio protein expression. Given the effects of these variants on TRIO GEF1/2 activities in vitro, we measured levels of active GTP-bound Rac1 and RhoA in brains of neonatal (P0) and young adult (P42) *Trio* variant-bearing mice (*Figure 1E–H*).

In concordance with the ~50% reduction in Trio levels in *+/K1918X* brains (*Figure 1C and D*), Rac1 activity was decreased in P0 *+/K1918X* brains (91% of WT activity, p<0.05; *Figure 1E*), with a trend toward decreased active RhoA levels (84% of WT, p=0.0865; *Figure 1F*). However, by P42, *+/K1918X* brains did not differ from WT in Rac1 or RhoA activity (*Figure 1G and H*), despite the persistent ~50% reduction in Trio9 protein level at this age (*Figure 1—figure supplement 1D–F and H–J*).

As K1431M decreases TRIO GEF1 catalytic activity in vitro (*Figure 1—figure supplement 1A and B*; *Sun et al., 2021*; *Katrancha et al., 2017*; *Sadybekov et al., 2017*), we anticipated Rac1 activity would also be reduced in *+/K1431M* mice. Instead, active Rac1 levels were significantly elevated in *+/K1431M* whole brain lysates compared to WT controls, showing a 111% increase at P0 (p<0.01;

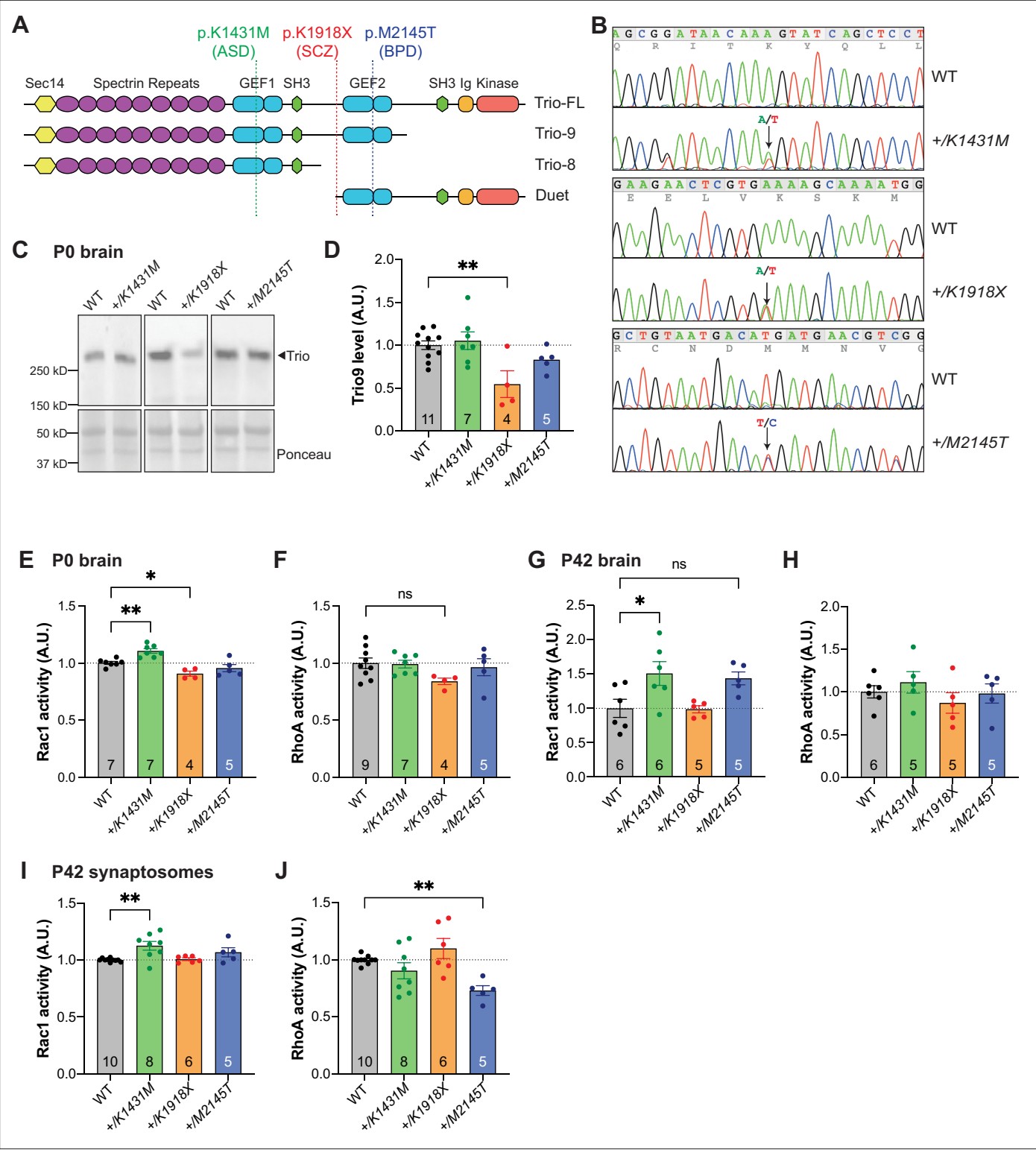

**Figure 1.** Genetically engineered mice with heterozygosity for K1431M, K1918X, or M2145T Trio variants have divergent effects on Trio protein expression and Rho GTPase activity. (**A**) Schematic of major Trio isoforms present in the adult mouse brain, with locations of engineered neurodevelopmental disease (NDD)-associated *Trio* variants: *K1431M* is a rare missense variant in the GEF1 DH domain associated with autism spectrum disorder (ASD); a *K1918X* nonsense variant that lies just before the GEF2 domain associated with schizophrenia (SCZ); and *M2145T* missense variant in the GEF2 DH domain found in an individual with bipolar disorder (BPD). (**B**) Representative sequencing chromatograms of WT and Trio variant mice. Arrows indicate heterozygosity for the variant alleles. (**C**) Representative immunoblots for Trio in P0 brain lysates using an antibody against Trio spectrin

*Figure 1 continued on next page*

*Figure 1 continued*

repeats (SR5-6). (**D**) Quantification of Trio protein levels from P0 brain lysates. Trio protein levels are reduced only in the brains of *+/K1918X* mice compared to WT controls (0.545±0.126 of WT level, p=0.0046). (**E–H**) Activity levels of Rac1 (**E,G**) and RhoA (**F,H**) in whole brain homogenates of neonate (P0, E–F) and adult (P42, G–H) *Trio* variant mice as measured by G-LISA assay. Rac1 activity is increased in *+/K1431M* mice relative to WT at both ages (1.106±0.027 fold at P0, p=0.0035; 1.509±0.175 fold at P42, p=0.0279) and decreased in neonate *+/K1918X* mice (0.908±0.0.032 fold, p=0.0230), with a trend towards increased activity in adult *+/M2145T* mice (1.438±0.183 fold, p=0.0843); meanwhile, RhoA activity appears unchanged in all mice relative to WT, although there may be a trend towards decreased activity in *+/K1918X* neonates (0.840±0.074 fold, p=0.1292). (**I,J**) Activity levels of Rac1 (**I**) and RhoA (**J**) in synaptosomes isolated from P42 mouse cortex. Rac1 activity is increased in *+/K1431M* synaptosomes (1.125±0.107 fold, p=0.0023), while RhoA activity is decreased in *+/M2145T* synaptosomes (0.731±0.042 fold, p=0.0093) relative to WT. All data shown as mean ± SEM. For (**D–J**), one-way ANOVA with post-hoc Bonferroni MC test identified differences from WT ($^{ns}p$ <0.1, *p<0.05, **p<0.01). Mouse numbers per group are shown in bars.

The online version of this article includes the following source data and figure supplement(s) for figure 1:

**Source data 1.** Full raw uncropped, unedited western blot files and Ponceau stains used for analysis displayed in *Figure 1C and D*.

**Source data 2.** PDF file containing annotated original western blot files and Ponceau stains for *Figure 1C and D*.

**Figure supplement 1.** Trio *+/K1918X* but not *+/K1431M* or *+/M2145T* mice have reduced levels of Trio protein in the brain.

**Figure supplement 1—source data 1.** Full raw uncropped, unedited western blot files and Ponceau stains used for analysis displayed in *Figure 1— figure supplement 1D–L*.

**Figure supplement 1—source data 2.** PDF file containing annotated original western blot files and Ponceau stains for *Figure 1—figure supplement 1D–L*.

**Figure supplement 1—source data 3.** Full raw uncropped, unedited western blot files and Ponceau stains used for analysis displayed in *Figure 1— figure supplement 1M–P*.

**Figure supplement 1—source data 4.** PDF file containing annotated original western blot files and Ponceau stains for *Figure 1—figure supplement 1M–P*.

*Figure 1E*) and an even greater increase of 150% at P42 (p<0.05; *Figure 1G*). Despite evidence of M2145T impacting TRIO GEF2 catalytic activity in vitro (*Katrancha et al., 2017*), we did not detect changes in active RhoA levels at the whole brain lysate level in *+/M2145T* mice at either age (*Figure 1F and H*).

Because Trio, Rac1, and RhoA are enriched at synapses (*O'Neil et al., 2021*; *Paskus et al., 2019*; *Katrancha et al., 2017*; *Oevel et al., 2024*), we also measured GTPase activities in synaptosomes (*Figure 1—figure supplement 1M–P*), within the same age range as the electrophysiological recordings performed on these mice. Active Rac1 was significantly increased in *+/K1431M* crude synaptosomes from P42 cortex (112% of WT, p<0.01; *Figure 1I*), consistent with our findings in *+/K1431M* whole brain lysates (*Figure 1E and G*). We also measured decreased levels of active RhoA in P42 *+/ M2145T* cortical synaptosomes compared to WT (73% of WT, p<0.01; *Figure 1J*), consistent with its reduced GEF2 activity in vitro (*Katrancha et al., 2017*; *Figure 1F and H*).

## Distinct trio variants differentially impact mouse behavior

NDDs affect learning and memory, compulsivity, motor coordination, and social skills; hence, we assessed these skills using a diverse array of established behavioral tests (*Katrancha et al., 2019*) in P42 mice bearing *Trio* variants (*Figure 2A*).

We measured motor coordination and learning using an accelerating rotarod test. *+/K1431M* and *+/K1918X* mice of both sexes fell from an accelerating rotarod with reduced latency relative to WT littermates, while *+/M2145T* mice performed similarly to WT (*Figure 2B*). In addition, *+/K1431M* and *+/K1918X* males and *+/K1431M* females showed a lower rate of improvement in this skill over repeated trials (*Figure 2B*). No deficits in muscle strength were noted in any genotype using the Kondziela inverted screen test prior to rotarod testing.

We used an open field test (OFT) and the elevated plus maze (EPM) to assess general locomotor activity and exploratory behavior. In the OFT (*Figure 2—figure supplement 1A–D*), *Trio* variant mice did not differ significantly from WT in total distance traveled or mean speed; however, *+/K1431M* females spent a greater amount of time in the outer and corner zones of the OFT compared to WT (*Figure 2—figure supplement 1D*, *right*), suggesting increased anxiety-like behavior in *+/K1431M* female mice. WT mice prefer the protected closed arms over the open arms in the EPM (*Figure 2— figure supplement 1E and F*), whereas *+/K1431M* and *+/K1918X* mice of both sexes and *+/M2145T* females did not exhibit this preference (*Figure 2—figure supplement 1F*).

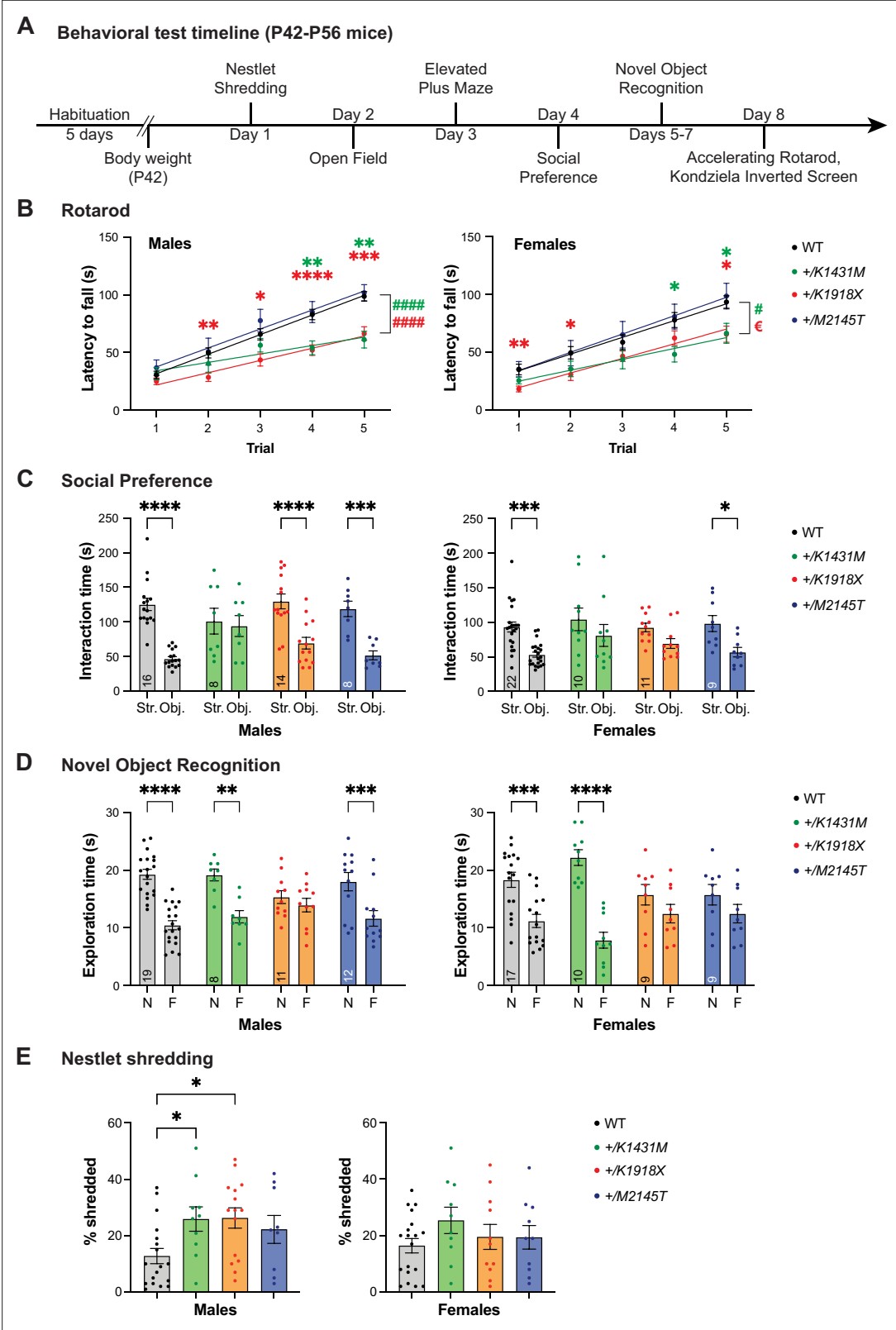

**Figure 2.** Heterozygosity for distinct *Trio* variants differentially impacts NDD-like mouse behaviors. (**A**) Schematic illustration of the behavioral tests performed on young adult (P42–P56) heterozygous *Trio* variant mice of both sexes. All mice proceeded through the same battery of tests. (**B**) *+/K1431M* and *+/K1918X* mice of both sexes had decreased latency to fall off an accelerating rotarod compared to WT male mice. In male mice (left), linear regressions identified differences from WT in slopes, indicating impaired rate of improvement in the skill (WT 16.96±1.344; *+/K1431M* 7.270±2.019,

*Figure 2 continued on next page*

*Figure 2 continued*

p<0.0001; *+/K1918X* 10.61 ± 1.444, p<0.0001; ####p<0.0001) (n=40 WT; 10 *+/K1431M*; 16 *+/K1918X*; 13 *+/M2145T* male mice). In female mice (right), linear regressions identified differences from WT in slopes (*+/K1431M* 9.436±2.146, p=0.0215; vs WT 14.52±1.792; #p<0.05) and intercepts (*+/K1918X* 6.492 ± 5.555, p=0.0248; vs WT 19.28±5.942; €p<0.05; n=28 WT; 11 *+/K1431M*; 16 13 *+/K1918X*; and 15 *+/M2145T* female mice). (C) *+/K1431M* mice of both sexes and *+/K1918X* females showed impaired social interactions in a three-chamber test, showing no preference to the (Str.) vs. inanimate object (Obj.) compared to WT. (D) *+/K1918X* mice of both sexes and *+/M2145T* females exhibit impaired novel object recognition and spend equal time exploring a novel object (N) and a familiar object (F). (E) Male *+/K1918X* mice exhibited increased nestlet shredding over 30 min (26.26 ± 3.61% shredded vs WT 14.26 ± 2.97%; p=0.0433), and *+/K1431M* mice exhibited a trend toward increased nestlet shredding (25.90 ± 4.34% shredded, p=0.1038) compared to WT mice. n=19 male, 19 female WT; 10 male, 10 female *+/K1431M*; 15 male, 11 female *+/K1918X*; 9 male, 10 female *+/M2145T* mice. All data are shown as mean ± SEM, significant differences identified using two-way ANOVA with post-hoc Bonferroni MC (ns p <0.1, *p<0.05, **p<0.01, ***p<0.001, ****p<0.0001). Numbers of mice quantified per group are annotated inside the bar unless otherwise indicated.

The online version of this article includes the following figure supplement(s) for figure 2:

**Figure supplement 1.** Heterozygosity for distinct Trio variants differentially impacts anxiety-like behaviors.

Deficits in social communication and social interaction are defining features of individuals diagnosed with NDDs (*American Psychiatric Association, 2013*). In a three-chamber social interaction test, *+/K1431M* mice of both sexes and *+/K1918X* females showed no preference for the stranger mouse over the object (*Figure 2C*), while *+/M2145T* mice of both sexes and *+/K1918X* males exhibited preference for the stranger mouse, similar to WT (*Figure 2C*). In a novel object recognition test, *+/K1918X* mice of both sexes and *+/M2145T* females failed to discriminate between the novel and familiar objects, while *+/K1431M* mice and *+/M2145T* males exhibited normal discrimination between novel and familiar objects similar to WT (*Figure 2D*).

Repetitive behaviors and stereotypies are often identified in individuals with NDDs (*Ridley, 1994*). We found a significant increase in compulsive nestled shredding in *+/K1918X* and *+/K1431M* males, with no significant changes in females (*Figure 2E*).

Behavioral phenotypes mostly overlapped between *+/K1431M* and *+/K1918X* mice, with differences in social interaction and memory tests. *+/M2145T* mice exhibited the fewest phenotypes in measured tasks. Additionally, some behavioral manifestations appeared to be selective to one sex. Together, these data indicate that these *Trio* alleles differentially impact behavior.

### *Trio +/K1431M* and *+/K1918X* mice have smaller brains

*TRIO* variants that reduce TRIO GEF1 activity are associated with microcephaly (*Barbosa et al., 2020*; *Bonnet et al., 2023*; *Bircher et al., 2022*), so we assessed head and brain size in *Trio* variant mice (*Figure 3A–E* [males], *Figure 3—figure supplement 1A–E* [females]). After adjusting for body weight, both the head width (*Figure 3D*, *Figure 3—figure supplement 1D*) and brain weight (*Figure 3E*, *Figure 3—figure supplement 1E*) were reduced in P42 adult *+/K1431M* mice of both sexes relative to WT, consistent with the microcephaly seen in patients harboring *TRIO* GEF1-deficient alleles. Of note, these ratios were driven by the increased body weight exhibited by P42 *+/K1431M* mice of both sexes compared to WT counterparts (*Figure 3C*, *Figure 3—figure supplement 1C*). *+/K1918X* mice of both sexes displayed a reduced brain weight-to-body weight ratio (*Figure 3E*, *Figure 3—figure supplement 1E*) but an unchanged head width-to-body weight ratio (*Figure 3D*, *Figure 3—figure supplement 1D*) relative to WT, suggesting a disproportionate reduction in brain mass in *+/K1918X* mice. Meanwhile, *+/M2145T* mice did not differ from WT in either head width or brain weight when adjusted for body weight.

### *Trio* variant mice show mild changes in cortical organization

We examined whether the decreased brain weight and head size observed in *+/K1431M* and *+/K1918X* mice were associated with anatomical defects. Gross histological analyses of fixed Nissl-stained brain sections in P42 male mice showed only mild morphological defects (*Figure 3F–J*). We observed reductions in both total cross-sectional brain area (*Figure 3F–G*) and cortical thickness (*Figure 3H–I*) in *+/K1918X* brains, consistent with their smaller head size and brain weight. Significant decreases in cortical layer 2/3 and layer 5 thickness were observed in *+/K1918X* brains (*Figure 3H–J*) and were in proportion to the relative decrease in cortical thickness relative to WT (*Figure 3—figure supplement 1F*). Meanwhile, no significant differences in brain area or cortical thickness were observed in either *+/K1431M* or *+/M2145T* mice.

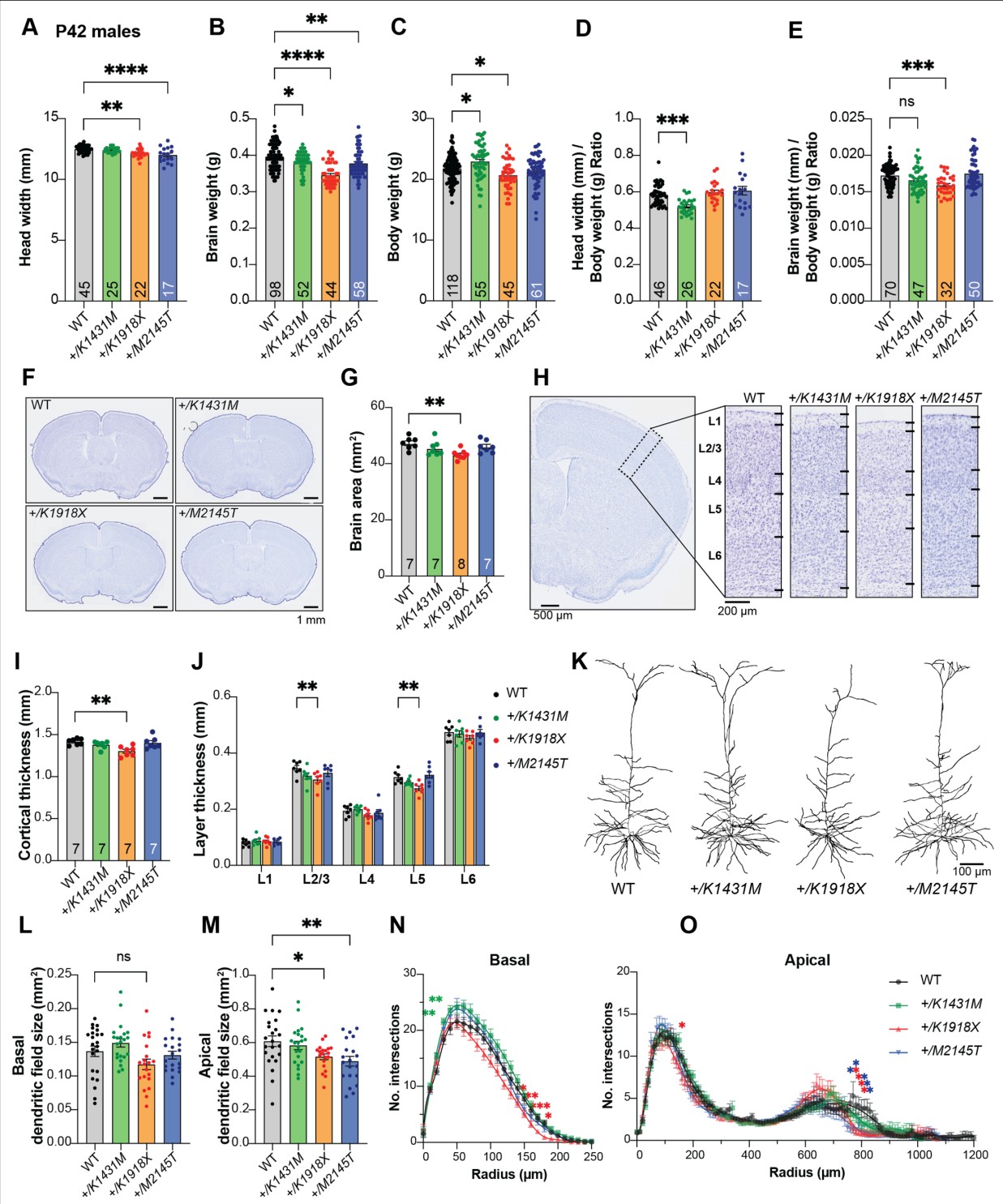

**Figure 3.** Trio +/K1431M and +/K1918X mice have smaller brain weights, but only +/K1918X brains have smaller, less complex neurons. (**A**) Ear-to-ear head width is reduced in P42 +/K1918X and +/M2145T compared to WT male mice (+/K1431M: 12.40±0.04 mm, p=0.95; +/K1918X: 12.15±0.08, p=0.001; +/M2145T 12.01±0.15, p<0.0001; vs WT 12.49±0.04 mm, n=17–45). (**B**) Brain weight is significantly decreased relative to WT in P42 males of all three heterozygous *Trio* variants (+/K1431M: 0.382±0.004 g, p=0.04; +/K1918X: 0.346±0.004 g, p<0.0001; +/M2145T 0.378±0.005 g, p=0.002; vs WT 0.396±0.004 g, n=44–98). (**C**) Body weight is significantly increased in P42 +/K1431M males and decreased in +/K1918X males (+/K1431M: 22.91±0.38 g, p=0.01; +/K1918X: 20.67±0.03 g, p=0.001; +/M2145T: 21.22±0.33 g, p=0.44; vs WT 21.76±0.19 g, n=45–118). (**D**) Head widths normalized to body weight of P42 +/K1431M male mice were reduced 10.8% compared to WT mice (+/K1431M: 0.520±0.008 mm/g, p=0.0001; +/K1918X: 0.598±0.012 mm/g, p>0.999; +/M2145T 0.607±0.023 mm/g, p=0.54; vs WT 21.76±0.19 mm/g, n=17–46). (**E**) Brain weights normalized to body weight of P42 +/K1431M

*Figure 3 continued on next page*

*Figure 3 continued*

and *+/K1918X* male mice were reduced 3.9% and 7.9%, respectively compared to WT mice (*+/K1431M*: 0.520±0.008 mm/g, p=0.0001; *+/K1918X*: 0.598±0.012 mm/g, p>0.999; *+/M2145T* 0.607±0.023 mm/g, p=0.54; vs WT 21.76±0.19 mm/g, n=17–46). (**F**) Representative images of Nissl-stained 30 μm coronal slices of male P42 WT and heterozygous *Trio* variant brains. (**G**) Total cross-sectional tissue area of Nissl-stained coronal sections was reduced by ~9% *+/K1918X* in P42 male mice compared to WT. (**H**) Representative images of Nissl-stained cortical layers (L1-L6, dotted black box) of P42 WT and heterozygous *Trio* variant brains. (**I**) The total cortical thickness (from H) is reduced by ~8% in *+/K1918X* P42 male brains compared to WT. (**J**) Thickness of individual cortical layers, as identified in Nissl stains in H. L2/3 and L5 were preferentially reduced (–12% and –13%, resp.) in *+/K1918X* cortex relative to WT (L2/3: 0.306±0.011 mm vs WT 0.346±0.010 mm, p=0.0043; L5: 0.274±0.008 mm vs WT 0.314±0.008 mm, p=0.0054). (**K**) Representative traces of M1 L5 PNs from heterozygous male *Trio* variant mice crossed with *Thy1-GFP(M)*. (**L**) *+/K1918X* M1 L5 PNs show a trend toward reduced basal dendritic field size (0.1172±0.0078 mm²; vs WT 0.1368±0.0077 mm², p=0.0933; n=15–22 neurons per mouse), as measured by convex hull analysis of dendrite arbor reconstructions. (**M**) Both *+/K1918X* and *+/M2145T* exhibit significantly smaller apical dendritic field size (*+/K1918X*: 0.5157±0.0169 mm², p=0.0460; *+/M2145T*: 0.4893±0.0285 mm², p=0.0062) compared to WT (0.6081±0.0319 mm²; n=15–22 neurons per mouse). All data shown as mean ± SEM. One-way ANOVA with post-hoc Bonferroni MC test identified significant differences from WT (^{ns}p <0.1, *p<0.05, **p<0.01). (**N,O**) Sholl analysis revealed basal (**N**) and apical (**O**) dendritic arborization changes in *Trio* variant M1 L5 PNs compared to WT: both basal and apical arborization was reduced in *+/K1918X*, while proximal basal arborization was increased in *+/K1431M*. Two-way ANOVA (stacked) with post-hoc Bonferroni MC test identified differences from WT.

The online version of this article includes the following figure supplement(s) for figure 3:

**Figure supplement 1.** Heterozygous *Trio* variant mice show mild alterations in cortical organization.

**Figure supplement 2.** Additional measurements of dendrites from M1 L5 PN reconstructions show modest order-dependent changes in *Trio* variant mice.

Deletion of *Trio* has been shown to impair the migration of forebrain interneuron progenitors, resulting in fewer neurons entering the cerebral cortex and altered distribution of cortical layers (*Sun et al., 2021*; *Wei et al., 2022*). However, we observed no change in total cortical cell density or in layer-specific cell density in heterozygous *Trio* variant male mice, nor were numbers of cortical NeuN +neuronal cells or PV +inhibitory neurons altered relative to WT, although there were trends toward increased DAPI + cell density in *+/K1918X* and increased NeuN + cell density in *+/M2145T* motor cortex (*Figure 3—figure supplement 1H–N*). These data suggest that the reduced brain size of *+/K1918X* mice results from a loss of neuropil rather than reductions in cortical neuron number.

## Trio variant heterozygotes exhibit alterations in dendritic arbors and synaptic ultrastructure

Altered dendritic arbor morphology and dendritic spine abnormalities are hallmarks of NDDs (*Kaufmann and Moser, 2000*; *Huttenlocher, 1970*; *Huttenlocher, 1974*; *Huttenlocher, 1991*; *Purpura, 1974*; *Purpura, 1975*; *Kulkarni and Firestein, 2012*). Excitatory neuron-specific ablation of one or both *Trio* alleles decreased dendritic arborization, increased spine density, and yielded smaller synapses in cortex area M1 Layer 5 pyramidal neurons (M1 L5 PNs; *Katrancha et al., 2019*).

Sholl analysis of M1 L5 PNs revealed significant reductions in both basal and apical dendritic arbor complexity and dendritic field size in P42 *+/K1918X* neurons, an increase in proximal basal arbor complexity in *+/K1431M* neurons, and a decrease in distal apical arbor complexity and apical dendritic field size in *+/M2145T* neurons relative to WT (*Figure 3K–O*, *Figure 3—figure supplement 1*).

Conditional *Trio* knockout specifically in forebrain excitatory neurons significantly increased the dendritic spine density on M1 L5 PNs, although they were smaller and appeared more immature (*Katrancha et al., 2019*). Notably, none of the *Trio* variant heterozygotes exhibited altered dendritic spine density on M1 L5 pyramidal neurons compared to WT mice on either apical or basal secondary arbors (*Figure 3—figure supplement 2L and M*).

Electron microscopy of cortical area M1 L5 revealed that synapse density was significantly increased in *+/K1918X* mice compared to WT (*Figure 4A and B*), possibly due to a net reduction in neuropil resulting from smaller dendritic arbors. Within synapses, postsynaptic density (PSD) length was slightly decreased in *+/K1918X* (by 6%) and *+/M2145T* (by 6.6%) mice relative to WT, but not in *+/K1431M* mice (*Figure 4C*). Cross-sectional presynaptic bouton area and spine head area were similar to WT in each *Trio* variant heterozygote (*Figure 4D and E*).

Synaptic vesicle (SV) distribution was significantly altered in *+/M2145T* mice relative to the other genotypes (*Figure 4F and G*). The densities of both docked and tethered vesicles at 15 and 50 nm from the active zone (AZ) respectively, which together can estimate the readily releasable pool (RRP)

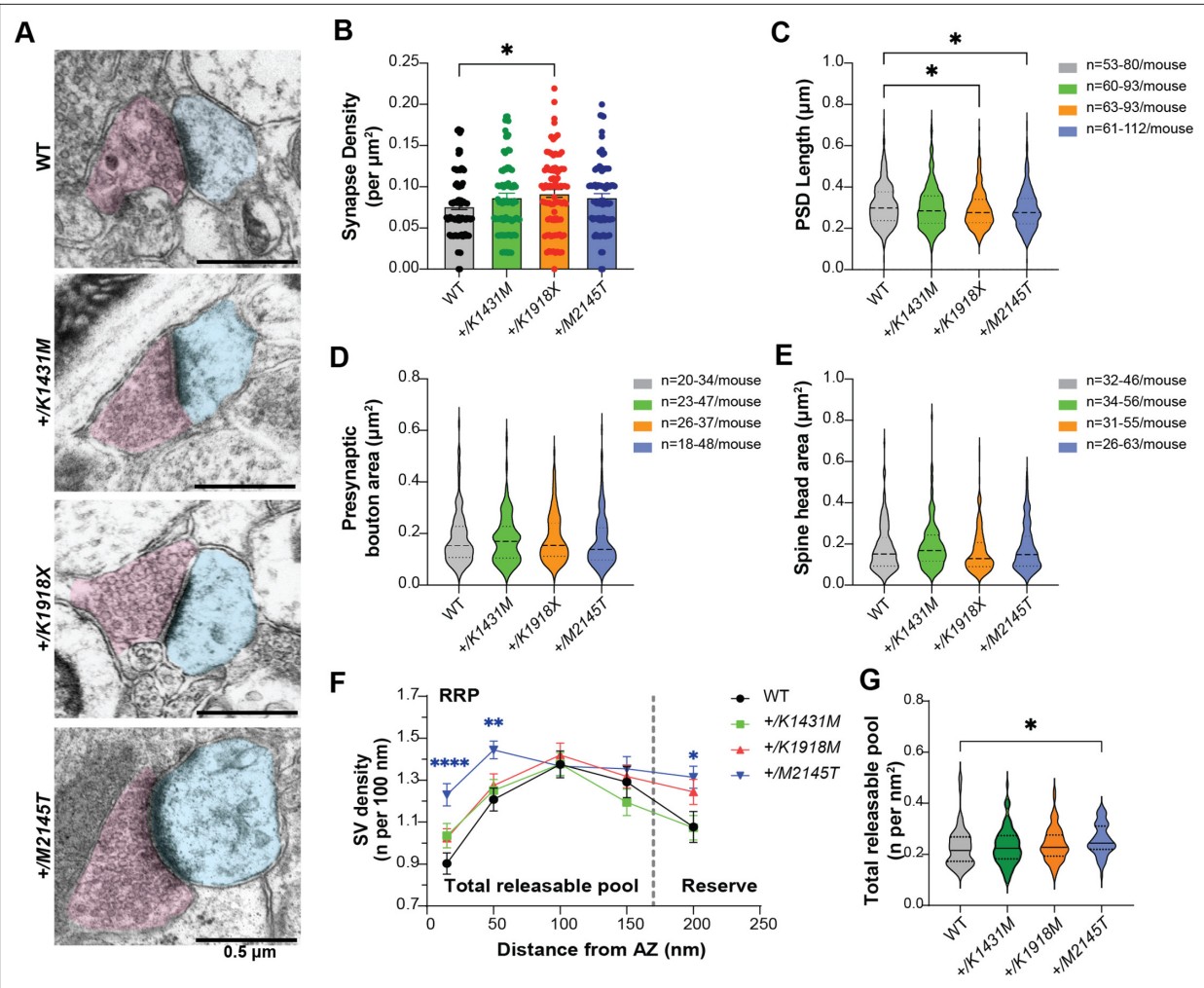

**Figure 4.** *Trio* variants differentially impact synapse ultrastructure and synaptic vesicle distribution. (**A**) Representative electron micrographs (EMs) from motor cortex layer 5 (M1 L5) of P42 WT and *Trio* variant mice. Post-synaptic regions are pseudo-colored in cyan; pre-synaptic regions in magenta. (**B**) Asymmetric synapse density was increased in *+/K1918X* mice (0.09205±0.004775 synapses/μm²; vs WT 0.07633±0.003954 synapses/μm², p=0.0345). (**C**) PSD lengths were slightly decreased in M1 L5 synapses by 6% in *+/K1918X* and 6.6% in *+/M2145T* mice vs WT (*+/K1918X* 0.2926 ± 0.004652 μm, p=0.0204; *+/M2145T* 0.2916±0.004922 μm, p=0.0142; vs WT 0.3125±0.005612 μm). (**D,E**) Presynaptic bouton and spine head areas of *Trio* variants M1 L5 synapses were unchanged from WT. (**F**) Synaptic vesicles (SVs) distribution per 100 nm of active zone (AZ) length in M1 L5 as a function of distance from the AZ. *+/M2145T* showed an increase in readily releasable pool (RRP) identified as docked SVs (15 nm from AZ; 1.23±0.05 vs WT 0.90±0.05) and increase in tethered SVs (50 nm from AZ; 1.44±0.04 vs WT 1.20±0.05). *+/K1918X and +/M2145T* also showed an increase in the reserve pool of SVs (200 nm from AZ; 3.51±0.21 and 3.81±0.18, resp. vs WT 2.74±0.16, n=15–30 synapses/mouse). (**G**) Total releasable pool, calculated as number of SVs at 15–150 nm from AZ per area of distribution (nm²). RRP (15–50 nm from AZ) was significantly increased in *+/M2145T* (0.257±0.007 vs WT 0.228±0.008), driven by increased docked and tethered SVs. All data are presented as mean ± SEM, significance tested by ordinary one-way ANOVA with post-hoc Bonferroni MC test (*p<0.05, **p<0.01, ***p<0.001, ****p<0.0001).

(*Montesinos et al., 2015*; *Dong et al., 2018*), were significantly increased in *+/M2145T* relative to the other genotypes (*Figure 4F and G*). Also, SV distribution at 200 nm from the AZ, thought to contribute the reserve pool, was also significantly increased in *+/M2145T* mice, suggesting an overall larger SV pool size. No differences in synaptic vesicle distribution were noted in *+/K1431M* mice, while *+/K1918X* showed modestly increased SVs at 200 nm from AZ relative to WT.

## Synaptic transmission and plasticity are differentially impaired by distinct *Trio* variants

Loss of Trio function or disruption of TRIO GEF1 activity in slice culture decreases AMPAR levels at excitatory synapses (*Katrancha et al., 2019*; *Tian et al., 2021*; *Paskus et al., 2019*; *Sadybekov et al.,*

*2017*), while mice bearing a GEF1-deficient *Trio* allele exhibited decreased gamma-aminobutyric acid receptor (GABAR)- and glycine receptor (GlyR)-mediated inhibitory miniature current (mIPSC) frequencies in the prefrontal cortex (*Sun et al., 2021*). To explore how *Trio* variants impact synaptic function, we measured both miniature excitatory currents (mEPSCs) and mIPSCs in M1 L5 PNs in each of the *Trio* variant heterozygotes at P35-42 consistent with other experiments.

AMPAR-mediated mEPSC amplitudes were significantly increased in *+/K1431M* and *+/K1918X* mice relative to WT littermates, with no change in their frequencies (*Figure 5A–C*). In contrast, AMPAR-mediated mEPSC amplitudes were unchanged in *+/M2145T* mice, but their frequencies were significantly increased. No significant changes in NMDAR mEPSC amplitudes were noted, while NMDAR mEPSC frequencies were decreased in *+/K1431M* and increased in *+/M2145T* mice cortex (*Figure 5D–F*). Notably, these findings correlated with gross alterations in the ratio of NMDAR/AMPAR-mediated evoked (e)EPSCs measured in M1 L5 PNs following stimulation in L2/3. *+/K1431M* and *+/K1918X Trio* variant heterozygotes showed decreased NMDA/AMPA ratios, indicating imbalances in NMDAR- versus AMPAR-mediated conductance (*Figure 5J and K*) and suggesting an increase in synaptic AMPAR signaling in both mice. Significant decreases in mIPSC frequencies were noted in *+/K1431M* and *+/M2145T* mice relative to WT mice, with no change in amplitudes (*Figure 5G–I*). *+/K1918X* mice exhibited increased mIPSC amplitude with no observed change in mIPSC frequency. Together, these data indicate that the *Trio* variants differentially impact excitatory and inhibitory transmission.

Finally, we tested the ability of the L5 PNs to undergo long-term potentiation (LTP) following theta-burst stimulation of L2/3 afferents (*Figure 5L and M*). While LTP was robustly induced and potentiated in M1 L5 PNs from WT mice, LTP induction and potentiation were deficient in slices from *+/K1918X* and *+/K1431M* mutant mice. In contrast, *+/M2145T* L5 PNs showed increased induction and prolonged potentiation of LTP compared to WT L5 PNs.

## Neurotransmitter release is altered in *Trio +/K1431M* and *+/M2145T* heterozygotes

In addition to its postsynaptic roles, Trio localizes presynaptically and interacts with the presynaptic active zone scaffolding proteins Bassoon and Piccolo (*Terry-Lorenzo et al., 2016*; *O'Neil et al., 2021*; *Paskus et al., 2019*). Additionally, recent work demonstrates that Rac1 activity levels can bidirectionally affect SV probability of release (Pr) in excitatory synapses (*O'Neil et al., 2021*; *Keine et al., 2022*; *Broadbelt et al., 2002*). To assess the impact of *Trio* variants on presynaptic function, we first measured the paired-pulse ratio (PPR) to test possible changes in Pr. Synapses with a low probability of release (Pr), such as L2/3 onto L5 PNs synapses, exhibit paired-pulse facilitation (PPF), where synaptic response increases for the second of two apposed stimuli due to elevated residual $Ca^{2+}$ promoting SV fusion. WT and *+/K1918X* M1 L5 PNs exhibited normal facilitation of eEPSC amplitudes that decreased with increased interstimulus interval (ISI) (*Figure 6A and B*). PPF was significantly enhanced in *+/K1431M* M1 L5 PNs at short ISIs relative to WT, suggesting a reduction in Pr at these synapses (*Figure 6A and B*). The *+/M2145T* PPR curve was complex, with significantly reduced PPF at short ISIs, yet clearly increased PPF at longer ISIs compared to WT (*Figure 6A and B*). The decreased PPF at initial ISIs in *+/M2145T* mice, together with an increase in both AMPAR and NMDAR mEPSC frequency (*Figure 5A–C*), demonstrates an increase in both synchronous and spontaneous glutamate Pr.

To further characterize changes in neurotransmitter release in these mice, we used high-frequency (40 Hz) stimulation (HFS) trains to quantitatively estimate glutamate Pr, RRP size, and rates of SV depletion and recovery. A plot of the normalized eEPSC responses to a HFS train stimulation again revealed facilitation upon the first 2–3 stimulations that was increased in *+/K1431M* and decreased in *+/M2145T* slices relative to WT, with no changes in *+/K1918X* slices (*Figure 6C and D*). Initial facilitation was followed by decaying eEPSC amplitudes, reflecting SV depletion under HFS. *+/K1431M* and *+/M2145T Trio* variants exhibited a slower train decay rate relative to WT during HFS with *+/M2145T* depleting at half the rate of WT ($\tau_d$, *+/K1431M:* 3.19 s, *+/M2145T:* 4.79 s vs WT: 2.70 s; *Figure 6D*).

We used a 'Decay' method (*Ruiz et al., 2011*; *Thanawala and Regehr, 2016*) to estimate Pr and RRP size from HFS trains, which allows us to account for initial facilitation seen in the train eEPSCs. Glutamate Pr in the L2/3-L5 synapses was increased in *+/M2145T* mice, while it was decreased for *+/K1431M* mice (*Figure 6E*), consistent with the relative changes observed in PPF for these mice. RRP

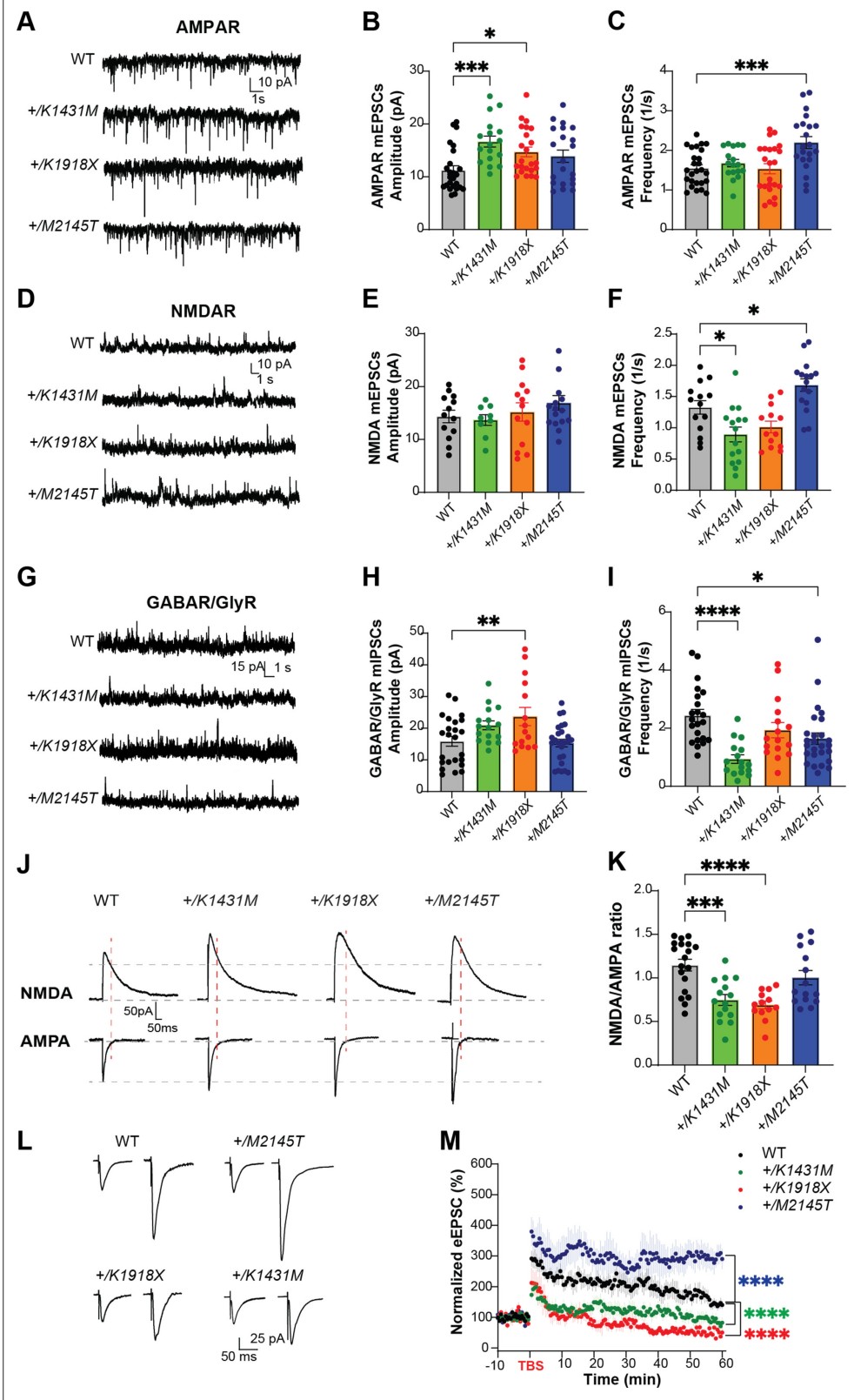

**Figure 5.** *Trio* variant mice exhibit deficits in synaptic signaling and LTP. (**A,D**) Representative traces of miniature excitatory AMPAR-mediated mEPSCs, NMDAR-mediated mEPSCs, and (**G**) inhibitory postsynaptic currents (mIPSCs) in M1 L5 pyramidal neurons of WT and Trio variant mice. (**B**) AMPAR-mediated mEPSC amplitudes were significantly increased in *+/K1431M* (16.67±1.04 pA; p=0.0009) and *+/K1918X* (14.71±0.92 pA; p=0.03) slices, with

*Figure 5 continued on next page*

*Figure 5 continued*

no observed changes in *+/M2145T* slices (13.90±1.16 pA; p=0.16) compared to WT (11.25±0.84 pA; n=17–25 neurons from ≥6–8 mice per group). (**C**) No significant changes in AMPAR mEPSC frequencies (q) were observed in *+/K1431M* and *+/K1918X*, while *+/M2145T* had an increase (2.20±0.15 1 /s; vs WT 1.55±0.09 1 /s; p=0.0005). (**E, F**) NMDAR mEPSC frequencies were reduced in *+/K1431M* (0.89±0.12 1 /s; vs WT 1.3324±0.11 1 /s; p=0.015) and showed an increase in *+/M2145T* mice (1.68±0.10 1 /s vs WT 1.3324±0.11 1 /s; p=0.044, n=9–13 neurons from ≥5–7 mice per group). (**H, I**) GABA/GlyR mIPSC amplitudes were significantly increased in *+/K1918X* vs WT (23.69±2.89 pA; vs 15.86±1.56 pA, respectably; p=0.008), while frequency was decreased in *+/K1431M* and *+/M2145T* (0.94±0.14 1 /s, p<0.0001; and 1.64±0.19 1 /s, p=0.013; respectably; vs WT 2.44±0.20; n=16–26 neurons from ≥6–8 mice per group). (**J**) Representative averaged traces of NMDA and AMPA eEPSCs recorded in M1 L5 PNs. (**K**) Heterozygous *+/K1431M* and *+/K1918X* Trio variants mice display reduced NMDAR/AMPAR eEPSC amplitude ratios, suggesting an increase in AMPAR transmission in M1 L5 PNs (*+/K1431M*: 0.75±0.06, p=0.0002; *+/K1918X*: 0.69±0.05, p<0.0001; *+/M2145T*: 1.00±0.08, p=0.37; vs WT: 1.15±0.07; n=13–19 neurons from ≥5–6 mice per group). (**L**) Averaged representative traces of baseline and post-TBS eEPSC currents in M1 L5 PNs of WT and *Trio* variant mice. (**M**) Normalized eEPSC amplitudes measuring LTP in L5 PNs by TBS in L2/3 afferents in all genotypes showed a significant decrease in the initiation and no potentiation of the LTP in *+/K1431M* and *+/K1918X*, with increase in initiation and potentiation of *+/M2145T* M1 L5 PNs compared to WT. LTP was induced at 0 min. RM two-way ANOVA with post-hoc Bonferroni MC test identified significant differences (n=6–8 neurons from ≥4–5 mice per group). Data are presented as mean ± SEM; significance tested by one-way ANOVA with post-hoc Bonferroni test unless specified otherwise (**p<0.01; ***p<0.001; ****p<0.0001).

size was much larger in L2/3-L5 synapses of *+/M2145T* mice relative to WT (*Figure 6F*), consistent with the increased SV distribution found at 15–50 nm from the AZ in electron micrographs (*Figure 4F and G*). We tested the ability of *Trio* variant heterozygotes to recover after train depletion by pairing HFS train with a single stimulus at increasing intervals (0.01, 2, 6, 9, 12, 18 s) and calculating the fractional recovery (see Materials and methods). The recovery rate ($\tau_R$) was significantly slower in *+/K1431M* L5 PNs, and they did not recover to their initial strength within 18 s, plateauing at 78% of maximal recovery compared to WT (*Figure 6G*). Together, *Trio +/K1431M* and *+/M2145T* mice exhibit distinct and significant alterations in short-term plasticity and synchronous glutamate release.

## *Trio* variant cortex displays different proteomic signatures

We used comparative proteomics from P21 cortex to identify proteins and pathways that were differentially altered by the *Trio* variants. We quantified a total of 7,362 proteins, finding distinct differences in the cortical proteome for each genotype (*Figure 7—figure supplement 1*, *Figure 7—figure supplement 1—source data 1*). Gene Set Enrichment Analysis (GSEA; *Subramanian et al., 2005*; *Mootha et al., 2003*) revealed alterations in distinct functions for each *Trio* variant (*Figure 7A*; *Figure 7—source data 1*). Of note, the only enriched gene set specific to neurons was downregulation of the synaptic vesicle pathway in *+/K1431M* cortex; all other gene sets were not cell-type specific.

We used SynGO (*Koopmans et al., 2019*) to investigate whether the *Trio* variants impacted synaptic functions. 1067 of the 7362 total quantified proteins were synaptic proteins listed in the SynGO gene set (*Figure 7*, *Figure 7—source data 2*). When restricted to brain-specific genes, all three *Trio* variant heterozygotes showed enrichment of differentially expressed proteins (DEPs) in synaptic processes (*Figure 7B and C*). Notably, *+/M2145T* upregulated DEPs and *+/K1431M* downregulated DEPs were significantly enriched at the presynapse (*Figure 7B*), but with only *+/K1431M* downregulated DEPs showing significant enrichment for postsynaptic receptor and synaptic vesicle cycling (*Figure 7C*). Meanwhile, *+/K1918X* showed enrichment of DEPs at the postsynapse, particularly in the postsynaptic density (*Figure 7B*). Together, our proteomics data point to a significant deficit in presynaptic function in both *+/K1431M* and *+/M2145T* cortex, as well as a significant effect of *+/K1431M* on postsynaptic function.

## Rho GEFs and synaptic regulatory proteins are altered in *Trio +/K1431M* and *+/M2145T* heterozygotes

Given our findings from proteomic analysis and electrophysiology, we measured levels of key presynaptic regulators, including synaptophysin (Syp), syntaxin binding protein 1 (Stxbp1, also known as Munc18-1), syntaxin 1a (Stx1a) and synaptotagmin 3 (Syt3), which are crucial for synaptic vesicle (SV) tethering, docking, replenishment, and calcium-dependent replenishment, respectively. We

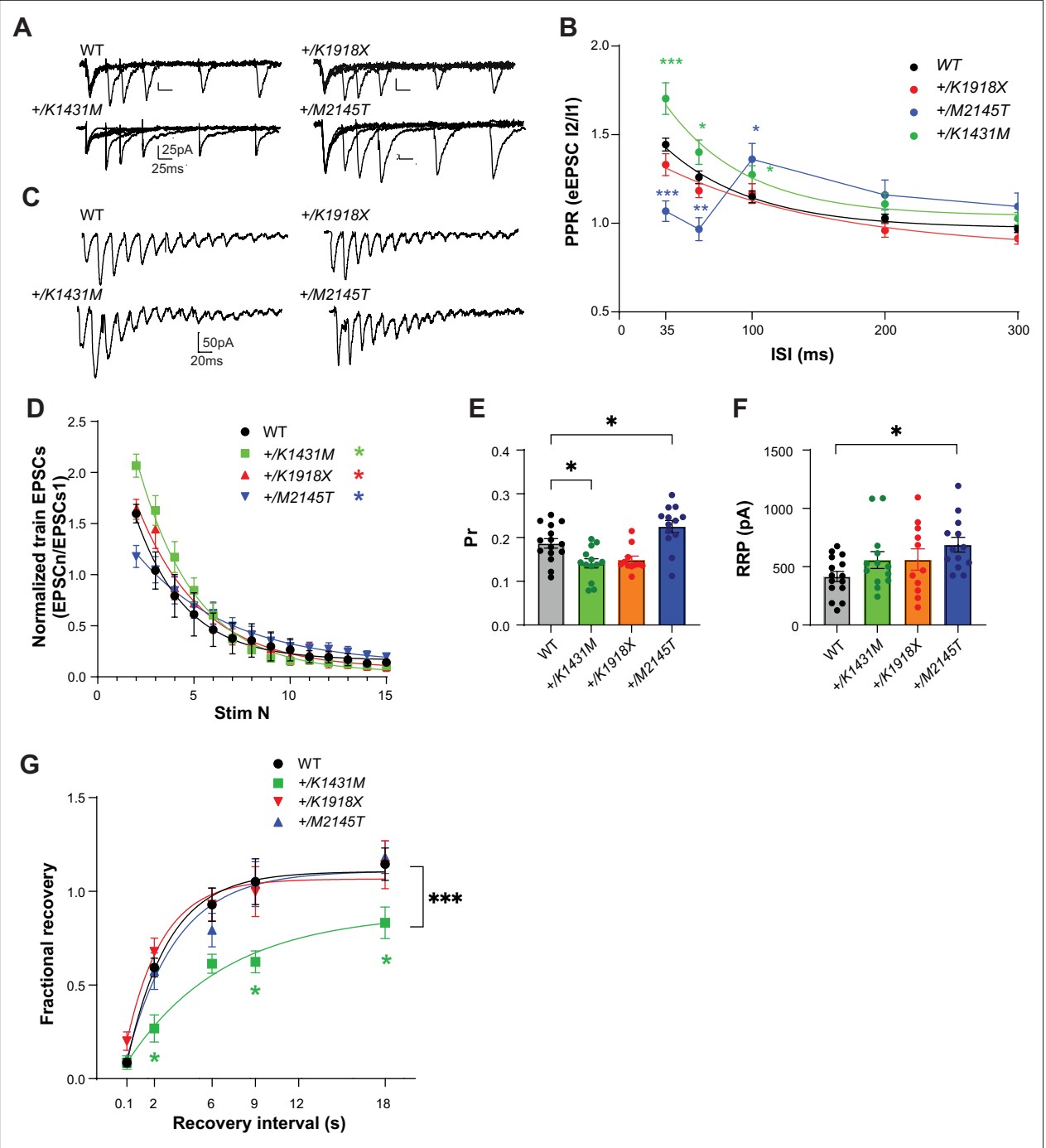

**Figure 6.** *Trio +/K1431M and +/M2145T* variant mice have deficiencies in short-term facilitation, glutamate Pr, and RRP. (**A**) Representative traces in M1 L5 PNs of WT, *Trio* variant mice in response to paired pulse stimulation in L2/3. (**B**) Paired-pulse ratio (PPR) at varying interstimulus intervals (ISIs) was overlaid with a single exponential fit (except for *+/M2145T* data). An increase in the initial PPR was observed in M1 L5 PNs of *+/K1431M* slices (35ms: 1.70±0.089, p=0.003; 60ms: 1.40±0.07, p=0.046; 100ms: 1.27±0.05, p=0.031; n=20–34 neurons from ≥7–9 mice per group) with no change in *+/K1918X* slices; and in *+/M2145T* slices we observed a decrease in initial RRP at shorter ISIs (35ms: 1.05±0.06, p<0.0001; 60ms: 0.97±0.06, p=0.037) and an increase at longer ISIs (100ms: 1.36±0.09, p=0.034; 200ms: 1.18±0.08, p=0.013) compared to WT (35ms: 1.40±0.04; 60ms: 1.21±0.03; 100ms: 1.13±0.03; 200ms 1.0±0.02; 300ms 0.96±0.17). (**C**) Representative traces of AMPAR eEPSCs in M1 L5 PNs under HFS (15 pulses at 40 Hz) in L2/3. (**D**) AMPAR $eEPSC_n$ amplitudes normalized to $eEPSC_1$ of the train revealed changes in the depletion rates during HFS in *Trio +/K1431M and +/M2145T* variants compared to WT (tau decay ($\tau_d$), WT: 2.7 s, *+/K1431M*: 3.19 s, *+/M2145T*: 4.79 s, *+/K1918X*: 2.52 s; n=12–15 neurons from 5 to 7 mice). (**E**) The estimated glutamate probability of release (Pr) was decreased in *+/K1431M* slices (0.13±0.099; p=0.013) and increased in *+/M2145T* slices (0.26±0.019, p=0.042), with no significant change in *+/K1918X* slices (0.15±0.01, p=0.64) compared to WT M1 L5 PNs (0.19±0.01; n=12–15 neurons from ≥5 mice per

*Figure 6 continued on next page*

*Figure 6 continued*

group). (**F**) The calculated size of the readily releasable vesicle pool (RRP) was increased only in *+/M2145T* M1 L5 PNs compared to WT (665.7±68.5 pA vs 415.8±43.9 pA, p=0.012). RRP in *+/K1431M* and *+/K1918X* synapses did not differ from WT (543.1±64.4 pA; and 543.1±64.4 pA, respectively vs 415.8±43.9 pA) (**G**) Exponential fits of the fractional recovery plotted vs ISI, to estimate synapse ability to recover from RRP depletion. Time of recovery, measured by exponential tau recovery ($\tau_R$), was significantly decreased in *+/K1431M* M1 L5 PNs (5.7 s, vs WT 2.2 s). *+/K1431M* also exhibited an inability to fully recover to initial levels after ISI 18 s, vs WT. Data are presented as mean ± SEM, with significant differences from WT tested using one-way ANOVA with post-hoc Bonferroni (*p<0.05; **p<0.01; ***p<0.001; ****p<0.0001).

also measured levels of presynaptic proteins in P42 cortical synaptosomes, where these proteins are enriched. Munc18-1, Syt3, and Syp levels were increased in *+/M2145T* synaptosomes relative to WT (*Figure 7D–H*). Meanwhile, Stx1a levels were significantly decreased in *+/K1431M* synaptosomes compared to WT, with no significant changes in *+/K1918X* compared to WT mice.

The elevated Rac1 activity in *+/K1431M* brain lysates and synaptosomes (*Figure 1I*) seemed at odds with previous reports that K1431M reduces TRIO GEF1 activity (*Figure 1—figure supplement 1A and B*; *Sun et al., 2021*; *Katrancha et al., 2017*; *Sadybekov et al., 2017*). We hypothesized that homeostatic compensation in *+/K1431M* mice may alter expression of other RhoGEFs and GAPs. Indeed, levels of the Rac1 GEF Tiam1 were increased in both *+/K1431M* and *+/M2145T* P42 cortical lysates, while VAV2 levels were increased in *+/M2145T* P42 lysates (*Figure 7I–L*). Levels of the Trio paralog Kalirin (*Yan et al., 2015*) were unaffected in the *Trio* variant mice at P42 (*Figure 7J*). Together, our proteomic analyses suggest that presynaptic functions are altered in *+/K1431M* and *+/M2145T* mice and may be driven by abnormal levels of crucial presynaptic regulatory proteins and changes in Rac1 and RhoA activity.

## NSC23766, a Rac1-specific inhibitor, rescues neurotransmitter release in *Trio +/K1431M* heterozygotes

Rac1 negatively regulates synaptic vesicle replenishment and synaptic strength in excitatory synapses (*O'Neil et al., 2021*; *Keine et al., 2022*). In *+/K1431M* synapses, increased Rac1 activity and decreased Stx1a levels were associated with reduced synaptic strength and impaired vesicle replenishment. We tested if the acute application of the Rac1 inhibitor NSC23766 (NSC) could rescue these deficits. Treatment of *+/K1431M* and WT slices acutely with NSC shifted the PPF downwards in M1 L5 PNs, suggesting an increase in Pr in both cases (*Figure 8A and B*). Notably, at longer ISI, PPRs in NSC-treated WT slices still normalized at around 1, while PPRs in NSC-treated *+/K1431M* slices shifted below 1, exhibiting slight depression.

Application of NSC to WT and *+/K1431M* slices under HFS train stimulation decreased initial facilitation in both (*Figure 8C and D*), and significantly increased Pr (*Figure 8E*) without affecting RRP size (*Figure 8F*) in both, rescuing the initial Pr in *+/K1431M* slices (*Figure 8C–F*). NSC treatment also led to a faster train decay rate ($\tau_d$) in *+/K1431M* slices but did not significantly change WT relative to untreated slices (*+/K1431M*+NSC $\tau_d$, 2.66 s vs. initial 3.19 s; WT +NSC: 2.85 vs 2.70 s; *Figure 8D*).

Finally, we tested if acute Rac1 inhibition impacts the rate of recovery of the RRP following HFS stimulation. NSC treatment increased the fractional recovery rate ($\tau_R$) in both WT and *+/K1431M* slices (WT +NSC: 32% faster than WT; *+/K1431M*+NSC: 40% faster than *+/K1431M*). NSC treatment of *the +/K1431M* variant allowed for full recovery at 18 s interval and sped up the recovery rate, but it remained significantly slower compared to WT (by 45%; *Figure 8G*).

Overall, we demonstrate that presynaptic Trio GEF1-dependent Rac1 signaling is crucial for maintaining synchronous glutamate Pr and SV replenishment at cortical L2/3-L5 synapses.

## Discussion

Large-scale genetic studies show significant overlap in risk genes for ASD, SCZ, and BPD, many converging on synaptic proteins (*Carroll and Owen, 2009*; *Purcell et al., 2014*; *Satterstrom et al., 2020*; *Genovese et al., 2016*; *Fromer et al., 2014*; *De Rubeis et al., 2014*; *Iossifov et al., 2014*; *Pinto et al., 2014*; *Kirov et al., 2012*). However, how variants in a single gene contribute to different NDDs remains a major unresolved question. Our study reveals that mice heterozygous for NDD-associated *Trio* variants differentially affecting Trio protein levels or GEF activity yield overlapping but distinct behavioral, neuroanatomical, and synaptic phenotypes. Our findings extend prior work

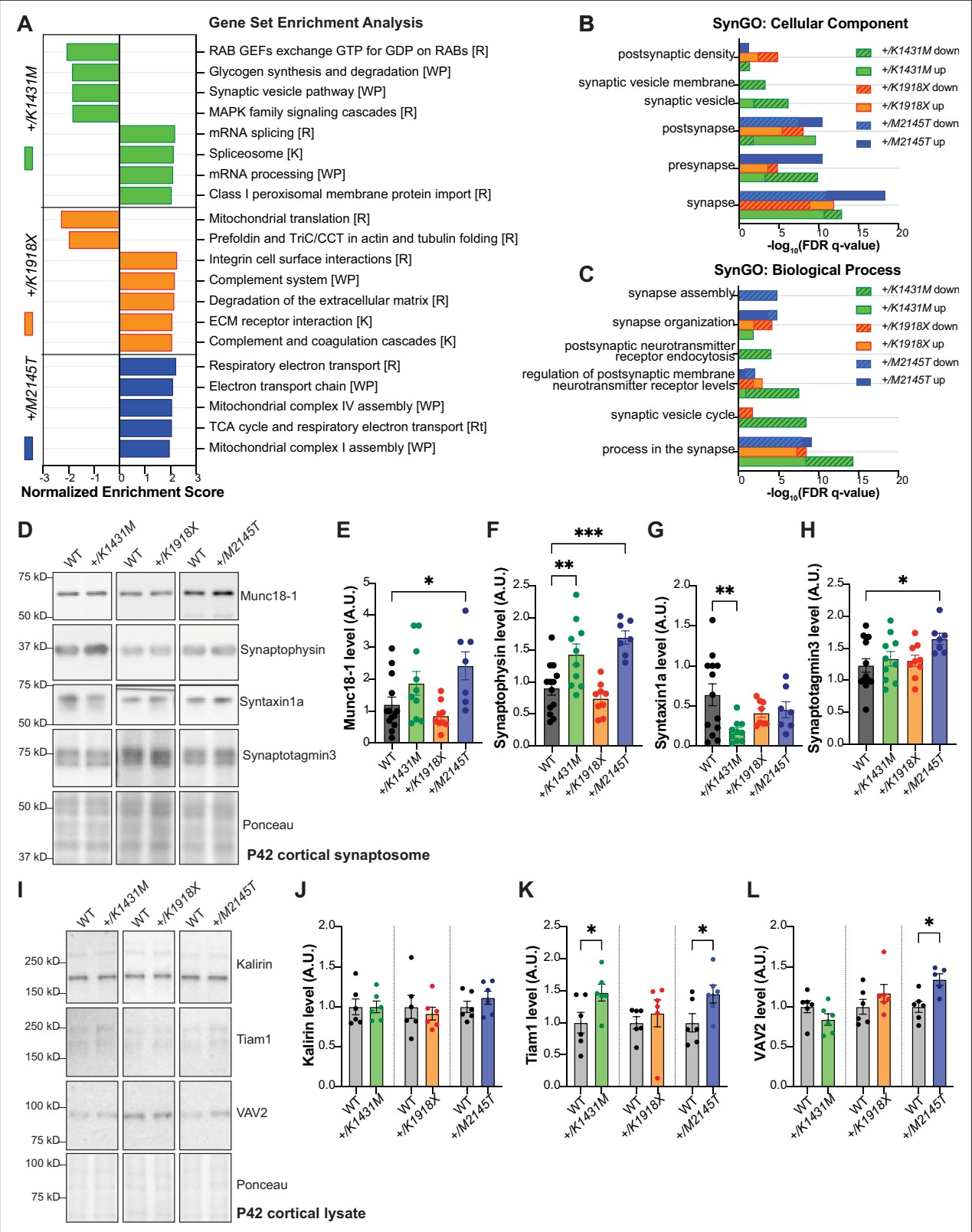

**Figure 7.** *Trio* variant mice show different molecular changes in the cortex involving presynaptic machinery and Rac1 GEFs. (**A**) Bar graph illustrating the top enriched pathways (FDR q-value <0.2, *FDR <0.05) identified by gene set enrichment analysis (GSEA) for each *Trio* mutant mouse compared to WT, using all 7,362 proteins quantified by mass spectrometry in P21 cortex (n=4/genotype), sorted by normalized enrichment score (NES). Pathways with +NES are upregulated, -NES are downregulated vs. WT. [ ] indicates gene set: [R] Reactome, [WP] WikiPathways, [K] KEGG. Full list attached in

*Figure 7 continued on next page*

*Figure 7 continued*

*Figure 7—source data 1*. (**B, C**) Bar graphs illustrating the top enriched (FDR q-value <0.001) (**B**) cellular components and (**C**) biological processes identified by GSEA, using synaptic proteins from SynGO gene sets (n=1,077 proteins), full list see in *Figure 7—source data 2*. (**D**) Representative immunoblots in synaptosomes isolated from P42 cortex of WT and *Trio* variant mice. (**E–H**) Normalized intensity levels from immunoblots demonstrate significant increases of (**E**) Munc18-1 (also known as syntaxin binding protein 1), (**F**) synaptophysin (Syp), (**G**) syntaxin1a (Stx1) and (**H**) synaptotagmin3 (Syt3) levels in *+/M2145T* synaptosomes; Syp is increased while Stx1a is significantly decreased in *+/K1431M* synaptosomes compared to WT. Ordinary one-way ANOVA with post-hoc Bonferroni MC test identified differences from WT (*p<0.05, **p<0.01, ***p<0.001; n=synaptosomes from 7 to 14 male mice). (**I**) Representative immunoblots of select RhoGEFs from P42 cortical lysates of WT and *Trio* variant mice. (**J–L**) Normalized intensity levels from immunoblots identified ~47% increase of Tiam1 levels in *+/K1431M* and increase ~45% in *+/M2145T* cortex vs to WT; VAV2 is increased by ~34% in *+/M2145T* cortex compared to WT. Unpaired t-tests identified differences from WT (*p<0.05; n=6 mice per genotype).

The online version of this article includes the following source data and figure supplement(s) for figure 7:

**Source data 1.** Gene Set Enrichment Analysis of P21 cortex proteome of *Trio* WT, *+/K1431M*, *+/K1918X*, and *+/M2145T* mice.

**Source data 2.** Synaptic Gene Ontologies (SynGO) of P21 cortex proteome of *Trio* WT, *+/K1431M*, *+/K1918X*, and *+/M2145T* mice.

**Source data 3.** Full raw uncropped, unedited western blot files and Ponceau stains used for analysis displayed in *Figure 7D–H*.

**Source data 4.** PDF file containing annotated original western blot files and Ponceau stains for *Figure 7D–H*.

**Source data 5.** Full raw uncropped, unedited western blot files and Ponceau stains used for analysis displayed in *Figure 7I–L*.

**Source data 6.** PDF file containing annotated original western blot files and Ponceau stains for *Figure 7I–L*.

**Figure supplement 1.** Mass spectrometry-based proteomics reveals molecular changes in the brains of *Trio* variant mice compared to WT mice.

**Figure supplement 1—source data 1.** Proteome of P21 cortex from *Trio* WT, *+/K1431M*, *+/K1918X*, and *+/M2145T* mice.

demonstrating that Trio is critical for postsynaptic signaling and synaptic plasticity. We also demonstrate for the first time in mice that Trio is critical for glutamate release and synaptic vesicle recycling, and that NDD-associated variants differentially impact these pre- and post-synaptic roles.

## Heterozygosity for *Trio* variants in mice yields phenotypes similar to those observed in NDDs

Individuals with mutations in *TRIO* present with a range of NDD-associated clinical features, including varying degrees of intellectual disability, altered head size, skeletal and facial features, and behavioral abnormalities (*Ba et al., 2016*; *Barbosa et al., 2020*; *Gazdagh et al., 2023*; *Pengelly et al., 2016*; *Bircher et al., 2022*; *Bonnet et al., 2023*; *Schultz-Rogers et al., 2020*). Patients with missense or truncating variants in *TRIO* that reduce GEF1 activity have mild developmental delay and microcephaly (*Barbosa et al., 2020*; *Bonnet et al., 2023*; *Pengelly et al., 2016*; *Bircher et al., 2022*). Similarly, we found that heterozygosity for the GEF1-deficient *K1431M* missense or the *K1918X* nonsense variants significantly reduced brain weight and/or head size compared to WT mice, along with multiple behavioral impairments. Notably, while both showed impaired motor coordination and learning, only mice bearing the ASD-associated *K1431M* allele exhibited social interaction deficits. In addition, we observed behavioral differences in male versus female *Trio* variant mice, possibly similar to human sex differences in the susceptibility to and clinical presentation of NDDs (*Bölte et al., 2023*).

Both *+/K1431M* and *+/K1918X* adult mice of both sexes had reduced brain-to-body weight ratios compared to WT, but these were driven by different factors. The smaller brain size in *+/K1918X* male mice was associated with a reduction in neuropil and reduced cortical thickness, similar to mice bearing excitatory neuron-specific ablation of one *Trio* allele (*Katrancha et al., 2019*) and paralleling the reduced gray matter volume and cortical thickness in SCZ patients (*Howes et al., 2023*; *Dabiri et al., 2022*; *Harvey et al., 1993*; *Suddath et al., 1989*; *Zipursky et al., 1992*). Meanwhile, *+/K1431M* mice of both sexes exhibited an overall increase in body weight leading to relatively decreased head width- and brain-to-body weight ratios in these mice. Adult *+/K1431M* male mice had no change in neuropil or cross-sectional brain area, consistent with a prior study describing normal brain size in *+/K1431M* at E14.5 (*Sun et al., 2021*). Rac1 mediates glucose-stimulated insulin secretion from pancreatic islet beta-cells (*Kowluru, 2011*; *Kowluru, 2021*; *Zhou et al., 2015*; *Veluthakal and Thurmond, 2021*; *Asahara et al., 2013*; *Sylow et al., 2016*), which may explain how chronic alterations in Rac1 activity contribute to weight changes in *Trio +/K1431M* and *+/K1918X* mice. Most studies of *Trio* variants have focused on neuronal effects, but expression of *Trio* in other tissues could explain the increased body weight in these mice, as well as the musculoskeletal abnormalities associated with *TRIO* variation in humans (*Ba et al., 2016*; *Barbosa et al., 2020*; *Gazdagh et al., 2023*; *Kloth et al., 2021*).

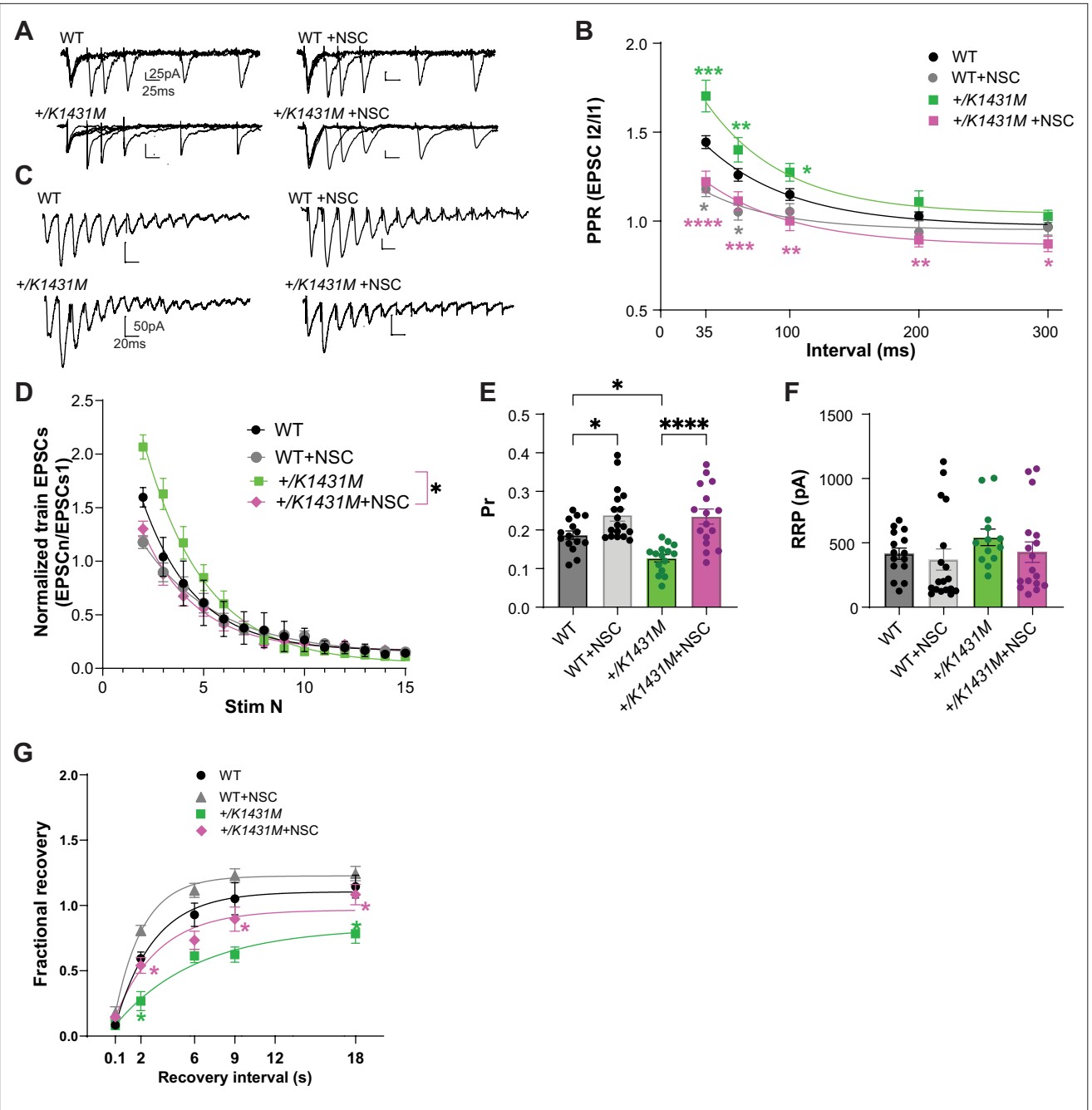

**Figure 8.** NSC, Rac1 inhibitor application rescued Pr in *+/K1431M* L23-L5 synapses and improved SV recycling. (**A**) Representative PPR traces of WT and *Trio +/K1431M* slices with or without 5 min application of 100 μM NSC23766. (**B**) Acute application of NSC onto both *+/K1431M* and WT synapses leads to a decrease in PPF in M1 L2/3-L5 synapses. *+/K1431*M slices significantly shifted the PPF curve at all ISI downwards compared to untreated *+/K1431M* slices and showed no significant difference from WT (*+/K1431M*+NSC 35ms: 1.25±0.06, p<0.0001; 60ms: 1.13±0.052, p=0.0007; 100ms: 1.02±0.053, p=0.0017; 200ms 0.91±0.039, p=0.0043; 300ms 0.88±0.045, p=0.021), with *+/K1431M* shifting into paired pulse depression at 200–300ms intervals, while WT PPF plateauing to 1. (**C**) Representative traces of AMPAR eEPSCs in M1 L5 PNs under HFS of WT and *Trio +/K1431M* slices before and after NSC application. (**D**) Normalized AMPAR eEPSC$_n$ amplitudes of the train revealed changes in the depletion rates during HFS before and after NSC application to WT and *+/K1431M* slices (tau decay ($\tau_d$), WT +NSC: 2.85 s vs WT: 2.70 s; *+/K1431M*+NSC: 2.66 s vs *+/K1431M*: 3.19 s, n=12–15 neurons from 5 to 7 mice). (**E**) Rac1 inhibition by NSC increased the glutamate Pr in both WT and *+/K1431M* slices (WT +NSC 0.25±0.067 vs initial 0.19±0.01, p=0.046; and for *+/K1431M*+NSC 0.23±0.019 vs initial 0.13±0.099, p<0.0001; n=15–18 neurons from ≥5 mice per group). (**F**) RRP in WT or *+/K1431M* synapses with NSC did not show significant changes to initial values (WT +NSC: 370.3±82.37 pA vs 415.8±43.9 pA, p>0.99; *+/K1431M*+NSC: 427.9±79.2 vs 543.1±64.44 pA, p>0.99). (**G**) Exponential fits of the fractional recovery for WT and *+/K1431M* with and without NSC application. NSC application led to a faster recovery time in WT (+NSC: 1.5 s vs initial 2.2 s) and it significantly improved but did not fully rescue recovery time in *+/K1431M* (+NSC

*Figure 8 continued on next page*

*Figure 8 continued*

3.2 s vs initial 5.7 s), but allowed for full recovery at 18 s. Data are presented as mean ± SEM, significance tested using one-way ANOVA with post-hoc Bonferroni (*p<0.05; **p<0.01; ***p<0.001; ****p<0.0001).

## *Trio* variants differentially impact dendritic arbor structure

Rac1 and RhoA signaling is critical for dendrite development (*Hall, 1998*; *Luo, 2000*; *Nakayama et al., 2000*; *Newey et al., 2005*; *Ba et al., 2013*). We found relatively subtle effects of Rac1/RhoA-altering *Trio* variants on cortical L5 PN dendrites. *+/K1918X* L5 pyramidal neurons are smaller and less complex than WT neurons, especially in the basal compartment corresponding to L5 where EM images were obtained, consistent with the smaller brain size and reduced cortical thickness of *+/K1918X* mice. We posit that due to their smaller dendritic field size, L5 neurons pack more densely, contributing to the mildly increased synapse density observed in *+/K1918X* M1 L5 cortex. Consistent with this hypothesis, we observed a trend toward increased DAPI + cell density in M1 L5 of *+/K1918X* neurons.

The reductions in dendritic arborization and length in *+/K1918X* neurons are consistent with reports of reduced gray matter volume and dendrite alterations in individuals with SCZ versus controls (*Broadbelt et al., 2002*; *Black et al., 2004*), but were modest compared to *NEX-Trio$^{+/fl}$* mice lacking one copy of Trio in excitatory neurons (*Katrancha et al., 2019*). Reduced Trio function in other cell types, such as in inhibitory neurons or glia, or in neurons from other brain regions that project to cortical M1 L5 PNs, may ameliorate the phenotypes in excitatory neurons of *+/K1918X* relative to *NEX-Trio$^{+/fl}$* mice. In addition, changes to L5 PN dendrites in *Trio* variant mice appeared to be regionally selective within the arbor: *+/K1431M* neurons had increased arborization only in proximal basal dendrites, while *+/M2145T* neurons had decreased arborization only in the most distal apical dendrites. These differences may reflect the differential spatiotemporal influence of Trio in regulating Rac1 versus RhoA activity. Alternatively, given our finding for presynaptic roles for Trio, these differences may reflect differential effects of the *Trio* variants on both excitatory and inhibitory afferent synaptic inputs, which play critical roles in shaping the apical and basal dendrites of L5 PNs (*Ramaswamy and Markram, 2015*).

## *Trio* variants impact brain Rho GTPase signaling

We show here that the *K1431M* variant significantly reduces TRIO GEF1 nucleotide exchange on Rac1 in vitro, consistent with previous reports (*Katrancha et al., 2017*; *Sadybekov et al., 2017*). Decreased Rac1 activity was observed in *+/K1431M* mice at embryonic day 14.5 in the ganglionic eminence, which is enriched for pre-migratory GABAergic interneurons (*Sun et al., 2021*). In contrast, we show an increase in active Rac1 levels in *+/K1431M* postnatal brains (at P0, P21, P42) and synaptosomes (at P42) from cortex, primarily composed of differentiated excitatory neurons. This increase aligns with observed phenotypes (e.g. reduced Pr) and reveals why reductions in brain volume, dendrites and spines, or AMPAR signaling anticipated from reduced Rac1 activity (*Chen et al., 2009*; *Reijnders et al., 2017*; *Tahirovic et al., 2010*; *Pennucci et al., 2019*; *Wiens et al., 2005*) were not observed in *+/K1431M* mice. We propose that the increased Rac1 activity we observed reflects homeostatic compensation in Rac1 regulation occurring between birth to adult ages and identify changes in Rac1-specific GEFs, for example Tiam1 and Vav2, that may contribute to this compensation. Importantly, we did not find changes in Kalirin levels in the adult brain of these Trio mutant mice, suggesting that Kalirin does not compensate for the loss of Trio GEF1 activity at this age.

We observed a significant reduction in active RhoA only in purified synaptosomes of *+/M2145T* brains, reflecting the synaptic compartment as a key locus of Trio function. In addition, despite *+/K1918X* mice having half the WT levels of Trio protein, we measured little to no change in active Rac1 and RhoA levels in *+/K1918X* brains. These findings are consistent with recent evidence that the spatiotemporally precise balance of Rac/Rho activity rather than absolute activity levels can mediate cytoskeletal rearrangements (*Grubisha et al., 2022*; *Duman et al., 2015*; *Ba and Nadif Kasri, 2017*; *Pertz, 2010*). Alterations in the activity of additional potential TRIO substrates, such as RhoG (*Bellanger et al., 2000*), Cdc42 (*Blangy et al., 2000*), and the neurodevelopmentally critical Rac3 (*Scala et al., 2021*; *Hajdo-Milasinović et al., 2007*; *Corbetta et al., 2009*), could also contribute to phenotypes in these mice.

Our results strongly complement a growing body of evidence showing that altered activation of Rho GTPases is an important mechanism affecting synaptic function in ASD (*Bölte et al., 2023*; *Pinto et al., 2010*; *Zeidán-Chuliá et al., 2013*; *Guo et al., 2020*; *Carbonell et al., 2023*; *Zamboni et al., 2018*). This highlights Rho GTPase signaling as a convergent pathway in ASD and therefore an attractive target for pharmacotherapy for these disorders.

## The NDD-associated *Trio* variants cause synaptic transmission, plasticity, and excitatory/inhibitory imbalance

Overexpression of a TRIO *K1431M* variant with reduced GEF1 activity decreased AMPAR-mediated mESPC amplitudes in rat organotypic slices (*Tian et al., 2021*), while Rac1 activation increased AMPAR amplitudes by promoting synaptic AMPAR clustering (*Wiens et al., 2005*; *Hussain et al., 2015*). In *+/K1431M* mice, we observed a decrease in the NMDA/AMPA ratio and NMDA mEPSCs frequency, with increased AMPAR mEPSCs amplitudes, with no changes in dendritic spines suggesting a possible reduction in silent synapses due to increased AMPAR incorporation. This increase in AMPAR was associated with increased active Rac1 levels measured in synaptosomes in adult *+/K1431M* mice. AMPAR mEPSC amplitudes were also increased in L5 PNs of *+/K1918X* mice, corresponding with a decrease in the NMDA/AMPAR ratio, but no changes in mEPSC frequencies, also suggesting an increased number of synaptic AMPARs. In *+/K1918X* cortex, Rac1 activity is reduced at P0 but increases by P21 and P42 to WT levels, suggesting that a *relative* increase in Rac1 during development may enhance AMPAR tone and affect LTP, especially given the precise spatiotemporal requirements of Rac/Rho activity and their effectors for LTP (*Wiens et al., 2005*; *Hussain et al., 2015*; *Cui et al., 2021*; *Herring and Nicoll, 2016*; *Duman et al., 2022*). The *+/M2145T* variant showed no changes in AMPAR or NMDAR amplitudes but had increased mEPSC frequencies in both, without changes in spine or synapse density, consistent with enhanced spontaneous neurotransmitter release.

mIPSC frequencies were decreased in *+/K1431M* and *+/M2145T* L5 PNs, while mIPSC amplitude was increased in *+/K1918X* slices. Sun et al. noted a similar deficit in inhibitory function in *+/K1431M* prefrontal cortex correlated with reduced interneuron migration to this region, including parvalbumin positive (PV+) neurons (*Sun et al., 2021*). In contrast, we did not observe reduced PV +interneuron numbers in the motor cortex in any *Trio* variant mice, suggesting that *Trio* variants may also impact the number of inhibitory synapses or transmission. However, we cannot fully exclude potential defects in the migration of other interneuron populations as contributors to these phenotypes.

Overall, heterozygosity for *Trio* variants dysregulates excitatory and inhibitory synaptic transmission in different patterns, resulting in E/I imbalance, a known driver of NDD phenotypes.

*Trio*-deficient excitatory neurons are unable to undergo long-term potentiation (LTP) in mouse brain slices (*Katrancha et al., 2019*), which is crucial for working memory in mammals. During LTP, Rac1 is suggested to be transiently activated and deactivated to regulate AMPAR (*Wiens et al., 2005*; *Hussain et al., 2015*; *Cui et al., 2021*; *Herring and Nicoll, 2016*; *Duman et al., 2022*). Both *+/K1431M* and *+/K1918X* L5 PNs exhibited reduced LTP induction and maintenance. The increased AMPAR resulting from elevated Rac1 activity (*+/K1431M* mice) or the inability of reduced levels of Trio to activate Rac1 (*+/K1918X* mice) may preclude LTP in these mice. *+/M2145T* mice showed a striking increase in the induction and maintenance of LTP, correlating with an increased glutamate Pr and SV pool size. The function of RhoA in plasticity is unknown, but the decrease in RhoA activity and increased levels in presynaptic machinery proteins measured in cortical synaptosomes may underlie the increase in LTP in these mice.

## *Trio* GEF1 and GEF2-deficient variants lead to opposing defects in the synaptic release of glutamate

In neuroendocrine, pancreatic beta, and mast cells, Trio GEF1 activity, Rac1, and RhoA are required for regulation of exocytosis (*Ferraro et al., 2007*; *Hong-Geller and Cerione, 2000*; *Lecuona et al., 2003*; *Momboisse et al., 2009*; *Pathak et al., 2012*). Recent work suggests that Trio GEF1 can act through Rac1 to regulate presynaptic processes – Rac1 colocalizes with SVs in the axonal boutons to negatively regulate action potential-dependent (synchronous) glutamate Pr and SV replenishment (*O'Neil et al., 2021*; *Keine et al., 2022*; *Doussau et al., 2000*; *Banerjee et al., 2024*). Using the Rac1 inhibitor NSC, we demonstrate that elevated Rac1 activity, potentially driven by Tiam1 upregulation, drives the reduction in synchronous glutamate Pr and SV replenishment without affecting RRP in +/

*K1431M* mice. NSC application rescues Pr and enhances SV replenishment in *+/K1431M* slices while also increasing both processes in WT. Decreased Stx1 levels in *+/K1431M* synapses, a key component of the priming machinery, may still be a limiting factor in effective priming, thus preventing full rescue of the SV recovery by NSC. Overall, NSC had a significant yet weaker effect in WT than in *+/K1431M* slices, likely due to the overall increased Rac1 activity in *+/K1431M* slices providing more of a target for NSC.

Conditional knockout of Rac1 in the Calyx of Held increased spontaneous (action potential-independent) glutamate Pr (*Keine et al., 2022*), aligning with the hypothesis that Rac1 activity negatively regulates neurotransmitter release by increasing the assembly of actin filaments at the AZ that impede SV fusion. *+/K1431M* cortical slices showed significant increases in AMPAR amplitudes, and our data are consistent with fewer silent synapses. Yet, we observed no change in AMPA mESPC frequency. We hypothesize that the Rac1-dependent decrease in Pr identified in these slices decreased AMPAR mEPSC frequency; hence, the opposing changes from these pre- and post-synaptic effects mask the expected change in AMPAR mEPSC frequency.

RhoA activity is significantly reduced in cortical synaptosomes of *+/M2145T* mice, and this is associated with increased RRP size. While a specific function for RhoA in regulating presynaptic release is currently unclear, levels of Munc18, Syp, and Syt3 are all increased in *+/M2145T* mice at P42 and can contribute to the enhanced Pr and altered SV cycling. The unusual PPR in the *+/M2145T* variant may result, at least in part, from altered Syt3 levels, crucial for calcium-dependent SV replenishment, as its deficiency causes similar phenotypes (*Weingarten et al., 2022*).

Our findings show that Trio GEF1 and GEF2-mediated Rho GTPase signaling pathways play critical and distinct roles in regulating Pr, RRP size, and SV recycling.

## Conclusions

*TRIO* is a risk gene for several NDDs with different patterns of variants observed in different disorders. We show here that variants in *Trio* that lead to impaired Trio levels or GEF function cause both shared and distinct defects in behavior, neuroanatomy, and prominent synaptic dysfunction that may reflect variant-specific NDD clinical phenotypes. Our data also demonstrate, for the first time, the differential impact of distinct *Trio* lesions on glutamate Pr, RRP size, and SVs replenishment, along with alterations in presynaptic release machinery that contribute to these deficits. We demonstrate that the *+/K1431M* lesion in TRIO GEF1 leads to presynaptic deficits due to Rac1 dysregulation, which can be rescued by pharmacological normalization. We propose that *Trio* variants disrupt brain function by impairing Rho GTPase signaling, causing diverse synaptic dysfunction through combined effects on pre- and/or post-synaptic functions. Our findings support growing evidence linking synaptic dysfunction in several NDD models to Rho GTPase signaling dysregulation, identifying it as a commonly affected pathway and a potential therapeutic target.

## Materials and methods

**Key resources table**

| Reagent type (species) or resource | Designation | Source or reference | Identifiers | Additional information |
|---|---|---|---|---|
| Strain, strain background (*Mus musculus*) | *Trio$^{+/+}$*(C57Bl/6) | Jax Laboratories (by Dr. CC Little) | RRID:IMSRJAX:000664 | Both sexes |
| Strain, strain background (*Mus musculus*) | *Trio$^{+/K1431M}$* | This paper | K1431M | Both sexes (see Generation of Trio mutant mice) |
| Strain, strain background (*Mus musculus*) | *Trio$^{+/K1918X}$* | This paper | K1918X | Both sexes (see Generation of Trio mutant mice) |
| Strain, strain background (*Mus musculus*) | *Trio$^{+/M2145T}$* | This paper | M2145T | Both sexes (see Generation of Trio mutant mice) |

*Continued on next page*

*Continued*

| Reagent type (species) or resource | Designation | Source or reference | Identifiers | Additional information |
|---|---|---|---|---|
| Sequence-based reagent | 5'-tgtaatacgactcactatagg ACCTGATCAAACCAGTTCA Ggttttagagctagaaatagc-3' | This paper | sgRNA scaffold sequence *K1431M* | see Generation of Trio mutant mice |
| Sequence-based reagent | 5'-tgtaatacgactcactatagg AAGCTTCTCACGCACGCA GGgttttagagctagaaatagc-3' | This paper | sgRNA scaffold sequence *K1918X* | see Generation of Trio mutant mice |
| Sequence-based reagent | 5'-tgtaatacgactcactatagg ATGACATGATGAACGTC GGGgttttagagctagaaatagc-3' | This paper | sgRNA scaffold sequence *M2145T* | see Generation of Trio mutant mice |
| Sequence-based reagent | 5'-gccaattccatctcttcctACCTGAT CAAACCTGTACAGCG TataacaaTgtatcagctcctttt aaaggtgtgtatgaaacattgtccatctgca acatcacaccctgtgttgatatgtatcccct-3' | This paper | Recombination template oligos *K1431M* | see Generation of Trio mutant mice |
| Sequence-based reagent | 5'-agctgccatctatggcaaaggcatttgta cacaggtgtgagagtgactcgcgaa ACTGCTGCATGCGTGAGAAGCT TaccattttgctttAcacgagttctt caatagcactgacgagctcagctgcgctgggag-3' | This paper | Recombination template oligos for *K1918X* | see Generation of Trio mutant mice |
| Sequence-based reagent | 5'-agggcccctcaataaacaatgcattacgtcaaatccctgca GTCGACCAacgttcatcG tgtcattacagcgcttcggcacgat gcacatgacttctacagctttctgtggg tggagagaaagca-3' | This paper | Recombination template oligos for *M2145T* | see Generation of Trio mutant mice |
| Sequence-based reagent | *primer pairs: GFP_For: 5'-gcacgacttcttcaagtccgccatgcc-3' GFP_Rev: 5'-gcggatcttgaagttcaccttgatgcc-3'* | This paper | Mice genotyping | see Generation of Trio mutant mice |
| Sequence-based reagent | *primer pairs: Trio1431_For: 5'-ttgtcattaatgtggtactgtgccc-3' Trio1431_Rev: 5'-gacaggccaagaaatgtcagtg-3'* | This paper | Mice genotyping | see Generation of Trio mutant mice |
| Sequence-based reagent | *primer pairs: Trio1918_ For: 5'-tacgagggagttcactgtctg-3' Trio1918_Rev: 5'-agtgcaggctatgcttcgttta-3'* | This paper | Mice genotyping | see Generation of Trio mutant mice |
| Sequence-based reagent | *primer pairs: Trio2145_For: 5'-gcctggacacatccgaattaga-3' Trio2145_Rev: 5'-aataaccccggacagaggaaag-3'* | This paper | Mice genotyping | see Generation of Trio mutant mice |
| Antibody | Rabbit anti-TRIO SR5-6 | **Katrancha et al., 2019** | | WB (1:1000) |
| Antibody | Rabbit anti-TRIO DH2 | **Katrancha et al., 2019** | | WB (1:1000) |
| Antibody | Mouse anti-PSD95 | NeuroMab | Cat#: K28/43 | WB (1:5000) |
| Antibody | anti-Synaptophysin (Syp; rabbit monoclonal) | Cell Signaling | Cat#: 36406 S | WB (1:5000) |
| Antibody | anti-Munc18-1/Stxbp-1 (rabbit monoclonal) | Abcam | Cat#: 109023 | WB (1:1000) |
| Antibody | anti-Syntaxin-1a (Stx1a; rabbit monoclonal) | Synaptic Systems | Cat#: 110 118 | WB (1:5000) |
| Antibody | anti-Synaptotagmin3 (Syt3; rabbit polyclonal) | Synaptic Systems | Cat#: 105 133 | WB (1:500) |
| Antibody | Rabbit anti-Kalirin (rabbit polyclonal) | **Yan et al., 2015** | (CT302) | WB (1:1000) |
| Antibody | anti-Tiam1 (rabbit monoclonal) | Cell Signaling | Cat#: 31128 | WB (1:1000) |

*Continued on next page*

*Continued*

| Reagent type (species) or resource | Designation | Source or reference | Identifiers | Additional information |
|---|---|---|---|---|
| Antibody | anti-VAV2 (rabbit recombinant monoclonal) | Abcam | Cat#: 52640 | WB (1:5000) |
| Antibody | anti-Rabbit IgG(H+L)-HRP conjugate (goat polyclonal) | Bio-Rad | Cat#: 170–6515 | WB (1:5000) |
| Antibody | anti-Mouse IgG(H+L)-HRP conjugate (goat polyclonal) | Bio-Rad | Cat#: 172–1011 | WB (1:5000) |
| Antibody | anti-NeuN (chicken polyclonal) | Synaptic Systems | Cat#: 266 006 | IF (1:2000) |
| Antibody | anti-Parvalbumin (PV; guinea pig monoclonal) | Swant | Cat#: GP72 | IF (1:1000) |
| Antibody | anti-Chicken IgY (H+L), AlexaFluor 488 (goat polyclonal) | ThermoFisher | Cat#: A-11039 | IF (1:2000) |
| Antibody | anti-Rabbit IgG(H+L), AlexaFluor 568 (goat polyclonal) | ThermoFisher | Cat#: A-11011 | IF (1:2000) |
| Commercial assay, kit | DAPI | Invitrogen | Cat#: D21490 | IF (1:10,000) |
| Chemical compound, drug | Kynurenic acid | Tocris Bioscience | Cat# 3694 | See Materials and Methods, Electrophysiology section |
| Chemical compound, drug | Lidocaine N-ethyl bromide (QX-314) | Tocris Bioscience | Cat# 2313 | See Materials and Methods, Electrophysiology section |
| Chemical compound, drug | D-2-Amino-5-phosphonovaleric acid (D-AP5) | Tocris Bioscience | Cat# 0106 | See Materials and Methods, Electrophysiology section |
| Chemical compound, drug | (-)-bicuculline methochloride (BMI) | Tocris Bioscience | Cat# 0109 | See Materials and Methods, Electrophysiology section |
| Chemical compound, drug | Strychnine hydrochloride | Tocris Bioscience | Cat# 2785 | See Materials and Methods, Electrophysiology section |
| Chemical compound, drug | Cyanquixaline (CNQX) | Tocris Bioscience | Cat# 0190 | See Materials and Methods, Electrophysiology section |
| Chemical compound, drug | NSC23766 | Tocris Bioscience | Cat# 2785 | See Materials and Methods, Electrophysiology section |
| Chemical compound, drug | Tetrodotoxin | Tocris Bioscience | Cat# 1078 | See Materials and Methods, Electrophysiology section |
| Recombinant DNA reagent (human) | WT TRIO GEF1 | *Blaise et al., 2022* | | Used for recombinant protein purification |
| Recombinant DNA reagent (human) | *K1431M* TRIO GEF1 | This paper | | Used for recombinant protein purification |
| Recombinant DNA reagent (human) | Rac1 | *Blaise et al., 2022* | | Used for recombinant protein purification |
| Sequence-based reagent | 5'-cagcgaataacgatgtatcagctcc-3' and 5'-ggagctgatacatcgttattcgctg-3'. | This paper | Oligonucleotides for site-directed mutagenesis (K1431M TRIO GEF1) | see In vitro GEF Assays |
| Commercial assay, kit | BODIPY-FL-GDP | Invitrogene | Cat# G22360 | See Materials and Methods |
| Commercial assay, kit | G-LISA activation assay kits for Rac1 | Cytoskeleton Inc. | Cat# BK128 | See Materials and Methods |
| Commercial assay, kit | G-LISA activation assay kits for RhoA | Cytoskeleton Inc. | Cat# BK124 | See Materials and Methods |
| Software, algorithm | AnyMaze software | Stoelting Co. | | Behavioral analysis |
| Software, algorithm | Fiji | https://fiji.sc/ | RRID:SCR_002285 | IF and WB analysis |
| Software, algorithm | Mini-Analysis software | Synaptosoft | | mEPSCs analysis |
| Software, algorithm | Origin Version 2021 | OriginLab | RRID:2JJ-JT7-8IP | Electrophysiology graphing, signal processing, and analysis |

*Continued on next page*

*Continued*

| Reagent type (species) or resource | Designation | Source or reference | Identifiers | Additional information |
|---|---|---|---|---|
| Software, algorithm | GraphPad Prism | Dotmatics | V10 | Statistical analysis and graphing |
| Software, algorithm | QuPath software | *Bankhead et al., 2017* | V5; https://qupath.github.io/ | Neuroanatomy analysis |
| Software, algorithm | pClamp software suite | Molecular Devices | 11.1 | Electrophysiology recordings, y graphing, signal processing, and analysis |

## Animal work

All animal work was performed in compliance with federal guidelines and approved by the Yale Institutional Animal Care and Use Committee. All mice were maintained on a C57Bl/6 background and housed on a standard 12 hr light/dark cycle. Mice heterozygous for *Trio* variants *K1431M* (*Trio$^{+/K1431M}$*), *K1918X* (*Trio$^{+/K1918X}$*), or *M2145T* (*Trio$^{+/M2145T}$*) were crossed with WT (*Trio$^{+/+}$*) mice to produce heterozygous *Trio* variant and WT littermates used in all experiments. As the mice were maintained on the same genetic background, analyses of WT mice were pooled and considered one genotype, irrespective of parental *Trio* variant genotype. Age-matched mice of both sexes were used for behavioral experiments and for brain and body weight measurements. Only male mice were used for electrophysiological and neuroanatomical analyses to reduce potential variation in dendritic spines and activity due to the estrus cycle (*Frankfurt and Luine, 2015*).

## Generation of *Trio* mutant mice

Mice heterozygous for *Trio* variants *K1431M*, *K1918X*, or *M2145T* were generated via CRISPR/Cas-mediated genome editing (*Yang et al., 2014*; *Chen et al., 2016*). Potential Cas9 target guide (protospacer) sequences in the vicinity of the *Trio K1431*, *K1918*, and *M2145* codons were screened using the online tool CRISPOR (*Haeussler et al., 2016*), and candidates were selected. Templates for sgRNA synthesis were generated by PCR from a pX330 template (Addgene), and sgRNAs were transcribed in vitro and purified (Megashortscript, MegaClear; Thermo Fisher). sgRNA/Cas9 RNPs were complexed and tested for activity by zygote electroporation, incubation of embryos to blastocyst stage, and genotype scoring of indel creation at the target sites. sgRNAs that demonstrated the highest activity were selected for creating the knock-in alleles. Guide primers for generating the template for transcription included a 5′ T7 promoter and a 3′ sgRNA scaffold sequence and were as follows (protospacer sequence capitalized):

*K1431M*: 5'-tgtaatacgactcactataggACCTGATCAAACCAGTTCAGgttttagagctagaaatagc-3'
*K1918X*: 5'-tgtaatacgactcactataggAAGCTTCTCACGCACGCAGGgttttagagctagaaatagc-3'
*M2145T*: 5'-tgtaatacgactcactataggATGACATGATGAACGTCGGGgttttagagctagaaatagc-3'

Recombination template oligos (Integrated DNA Technologies, San Diego, CA) were designed to create the desired codon changes, with incorporation of silent mutations to destroy the PAM and prevent sgRNA recognition of the newly created alleles as well as create a new restriction site for genotyping. sgRNA/Cas9 RNP and the template oligo were electroporated into C57Bl/6 J (JAX) zygotes (*Chen et al., 2016*). Embryos were transferred to the oviducts of pseudopregnant CD-1 foster females using standard techniques (*Nagy, 2003*). Genotype screening of tissue biopsies from founder pups was performed by PCR amplification and Sanger sequencing to identify the desired base changes, followed by backcrossing to C57Bl/6 mice and sequence confirmation to establish germline transmission of the correctly targeted alleles. Recombination template oligos were as follows (silent mutations bolded and italicized; new restriction digest site underlined):

*K1431M*: 5′-gccaattccatctcttcctACCTGATCAAACC*TGTACA*GCG*T*ataacaa*T*gtatcagctcctttta aggtgtgtatgaaacattgtccatctgcaacatcacaccctgtgttgatatgtatcccct-3′ (creates BsrGI site)
*K1918X*: 5′-agctgccatctatggcaaaggcatttgtacacaggtgtgagagtgactcgcgaaACT*GCTGC*ATGC GTGAGAAGCTT*accattttgctttA*cacgagttcttcaatagcactgacgagctcagctgcgctgggag-3′ (creates SphI site)

*M2145T:* 5'-agggcccctcaataaacaatgcattacgtcaaatccctgca<u>GTCGAC</u>CAacgttcatcGtgtcattacagc gcttcggcacgatgcacatgacttctacagctttctgtgggtggagagaaagca-3' (creates SalI site)

## Mouse genotyping

Genotypes were determined by PCR of mouse DNA using the following primer pairs:

*GFP_For*: 5'-gcacgacttcttcaagtccgccatgcc-3'
*GFP_Rev*: 5'-gcggatcttgaagttcaccttgatgcc-3'
*Trio1431_For*: 5'-ttgtcattaatgtggtactgtgccc-3'
*Trio1431_Rev*: 5'-gacaggccaagaaatgtcagtg-3'
*Trio1918_For*: 5'-tacgagggagttcactgtctg-3'
*Trio1918_Rev*: 5'-agtgcaggctatgcttcgttta-3'
*Trio2145_For*: 5'-gcctggacacatccgaattaga-3'
*Trio2145_Rev*: 5'-aataaccccggacagaggaaag-3'

To distinguish *Trio* variant and WT alleles, PCR was followed by restriction digest at 37 °C for at least 4 hours (BsrGI for *K1431M*, SphI for *K1918X*, SalI for *M2145T*) and subjected to agarose gel electrophoresis.

WT samples resulted in a band at 530 kb for *K1431* litters, 550 kb for *K1918* litters, 400 kb for *M2145* litters; while *Trio* heterozygous variants resulted in bands at 450 and 530 kb for +/*K1431M*, 350 and 550 kb for +/*K1918X*, 350 and 400 kb for +/*M2145T*. All mice used in experiments were genotyped twice from DNA samples collected at two different time points, prior to weaning and again post-experiment for validation.

## In vitro GEF assays

The *K1431M* point mutant was generated via site-directed mutagenesis of human WT TRIO GEF1 using the following oligos: 5'-cagcgaataacgatgtatcagctcc-3' and 5'-ggagctgatacatcgttattcgctg-3'. Recombinant WT and *K1431M* human TRIO GEF1 and Rac1 proteins were purified from bacteria as previously described (*Blaise et al., 2022*). GEF activity was monitored by the decrease in fluorescent signal ($\lambda_{excitation}$ = 488 nm; $\lambda_{emission}$ = 535 nm) as GTP was exchanged for BODIPY-FL-GDP on Rac1 over 30 min, as previously described (*Bircher et al., 2022*; *Blaise et al., 2022*).

## Body weight, brain weight, and head width measurements

Ear-to-ear head width of P42 mice was measured with calipers in anesthetized mice. Brain weight was measured after removing the olfactory bulbs and brain stem.

## Brain lysate preparation

Whole brain lysates from P0-P1 or P35-P42 mice were prepared as previously described (*Katrancha et al., 2019*; *Hollingsworth et al., 1985*; *Scheetz et al., 2000*), with minor modifications. Cerebella, hippocampi, and cortices were rapidly removed, snap-frozen in liquid nitrogen and stored at –80 °C until lysate preparation. Tissue was homogenized in ice-cold RIPA buffer (1% NP-40, 0.1% SDS, 50 mM Tris pH 8, 150 mM NaCl, 0.5% sodium deoxycholate) supplemented with protease and phosphatase inhibitors (Roche 11873580001; Roche 04906837001), then clarified by brief centrifugation; aliquots were snap frozen and stored at –80 ° C until immunoblotting. For Trio immunoblots, tissue was sonicated in homogenization buffer (1% SDS, 50 mM Tris pH 7.4, 2 mM EDTA), supplemented with protease and phosphatase inhibitors, heated at 95 ° C for 5 min, then clarified, snap frozen, and stored at –80 ° C until immunoblotting. Protein concentrations were determined with a bicinchoninic acid (BCA) assay (Pierce).

## Cortical synaptosome prep

Crude synaptosomes were prepared as previously described (*Hollingsworth et al., 1985*; *Scheetz et al., 2000*). Briefly, cortices were rapidly dissected from anesthetized P39-P42 mice and immediately homogenized in cold buffer (118 mM NaCl, 4.7 mM KCl, 1.2 mM $MgSO_4$, 2.5 mM $CaCl_2$, 1.53 mM $KH_2PO_4$, 212.7 mM glucose) supplemented with protease and phosphatase inhibitors (Roche). Homogenates were passed through a series of nylon filters of descending pore size: 40 µm, 10 µm, and finally 5 µm. Samples were centrifuged for 15 min at 1000 x *g* at 4 °C; the supernatant was discarded, and

the pellet was resuspended in an appropriate buffer for G-LISA measurements or immunoblotting. Mice from at least 3 litters per genotype were processed concurrently with WT littermates.

## Western blot and quantification

Brain lysates were separated on SDS-PAGE gels, transferred to nitrocellulose membranes, stained with Ponceau S, blocked in 5% nonfat milk in TBS-T, and incubated with primary antibodies overnight at 4 °C, then with conjugated secondary antibodies at RT for 1 hr (see **Key Resources Table** for list of antibodies). Images were captured by a ChemiDoc Imaging System (Bio-Rad) and quantified in ImageJ. Signal intensity was normalized to Ponceau S, then to the WT average.

## G-LISA

Active GTP-bound GTPase levels in brain lysates were measured using G-LISA activation assay kits for Rac1 (Cytoskeleton, Inc, BK128) and RhoA (Cytoskeleton, Inc, BK124). Brains or crude synaptosome pellets were homogenized in G-LISA lysis buffer supplemented with protease and phosphatase inhibitors (Roche), normalized to the same concentration as determined by Precision Red Advanced Protein Assay, and applied to the G-LISA matrices and processed according to manufacturer's protocols. Absorbances at 490 nm ($OD_{490}$) were background-subtracted and normalized to the WT average.

## Behavioral tests

Behavioral tests were performed in both male and female mice 6–8 weeks of age (P42-P56). Mice were habituated to handling for 5 min/ day for 5 days prior to experiments, and habituated in the test facility separate from housing for at least 30 min prior to starting the task. We performed the Kondziela inverted screen test for motor strength, accelerating rotarod for motor coordination and learning, social preference test, novel object recognition task, open field test, elevated plus maze, and nestlet shredding, as previously described (*Katrancha et al., 2019*; *Deacon, 2013*; *Omar et al., 2017*; *Rapanelli et al., 2017*; *Sfakianos et al., 2007*). All behavioral studies were performed and analyzed by an experimenter blinded to genotype. Individual data points could be excluded from analysis if a mouse failed to properly acclimate to the testing conditions, as described in detail for specific tests below.

## Accelerating rotarod

A five-lane rotarod treadmill (Med Associates ENV-577M 8.75" rod circumference) was used for accelerating analyses. Mice were acclimated to the rotating rod at 4 RPM for 10 s before beginning the test; mice that were unable to stay on the rod during this period were excluded from analyses. The rod then accelerated from 4 to 40 RPM over the course of 5 min (1.2 RPM/10 s) before leveling off at 40 RPM for 2.5 min. Five trials were performed per mouse with a rest time of at least 10 min between trials. The latency to fall was recorded per trial.

## Kondziela inverted screen test

The screen apparatus consisted of a 40 cm square of wire mesh composed of 12 mm squares with 1 mm diameter wire, surrounded by a square wooden frame 3.5 cm thick. Mice were placed in the center of the screen and inverted over 2 s over a clear open box 45 cm high. During the training session on day 1, mice were habituated to hanging on the screen until a cumulative hang time of 2 min was reached; mice were excluded if they were unable to meet this requirement. During the test session on day 2, mice were placed on the screen as before and given three trials to reach a maximum inverted hang time of 8 min, with 10 min rest periods between trials. Mice that were unsuccessful at reaching the 8 min mark were excluded from the accelerating rotarod test.

## Open field test

Mice were placed in a large 16x16 inch square clear plexiglass SuperFlex cage and monitored using the Fusion software connected to a light beam array (Omnitech Electronics, Open Field Test, Version 4.5) for 10 min. The following measurements were analyzed based upon beam breaks: total distance traveled, ambulatory time (successive beam breaks), and vertical activity time (z axis beam breaks).

## Social preference test

Mice were placed in a 50 cm x 50 cm plexiglass box with two wire-mesh pencil holders (open side facing downward) for 10 min for habituation. Then, the mouse was briefly removed while an inanimate object (Duplo blocks of a similar size and color to a mouse) and a male conspecific "stranger" mouse were placed underneath separate wire-mesh pencil holders. The test mouse was then returned to the cage for 10 min, tracked using the AnyMaze software (Stoelting Co.), and scored for entries into and time within a social target zone and an equivalent nonsocial target zone. The target zones were defined by an annulus around the wire enclosure.

## Novel object recognition task

During object familiarization on day 1, two identical objects were placed on the left and right sides of the testing cage. Mice explored the objects until they accumulated 30 s of tactile exploration time, defined as direct oral or nasal contact. Mice were then returned to their home cage for 48 hr. During the novel object recognition task on day 3, one familiar and one novel object were placed on either side of the testing cage, and mice explored the objects until they accumulated 30 s of tactile exploration time. Mice were excluded from analysis if they failed to explore both objects or accumulate 30 s of tactile exploration time within 6 min on either day.

## Elevated plus maze

The apparatus is raised 36.5 cm above the ground with a 5 cm square center platform and four 30 cm x 5 cm arms; two of the arms are enclosed by 16 cm high walls (Stoelting Co, Wood Dale, IL). Mice were tested in 3 trials that were 5 min long with at least 10 min to rest between trials. Mice were placed directly on the center platform and allowed to freely explore all arms of the maze, tracked using the AnyMaze software (Stoelting Co.), and scored for time spent in each arm.

## Nestlet shredding

A standard polycarbonate mouse cage (19×29 × 13 cm) was filled with unscented bedding material to a depth of 0.5 cm, the surface was leveled, and a fitted filter-top cover was placed on top. Commercially available cotton fiber nestlets (5 cm x 5 cm, 5 mm thick, 2–2.5 g each) were weighed on an analytical balance before being placed on top of the bedding. The mouse was placed into the test cage without food or water and left undisturbed for 30 min. The remaining intact nestlet was removed and allowed to dry overnight. The weight of the dried unshredded nestlet was divided by the starting weight to calculate the percentage of nestlet shredded.

## Animal perfusion and tissue processing

Mice were transcardially perfused with heparinized PBS followed by 4% paraformaldehyde (PFA) in phosphate buffered saline (PBS). Brains were postfixed in 4% PFA at 4 °C for 24 hr, then sliced coronally at 30 µm for Nissl stain and immunohistochemistry and at 200 µm for dendritic arbor and spine analysis. Sections were processed as per application and coverslip-mounted with ProLong Diamond Antifade Mountant (P36961, Thermo Fisher). For electron microscopy, mice were perfused with 4% PFA supplemented with 2% glutaraldehyde.

## Nissl staining

Nissl staining on 30 µm brain slices was performed according to standard protocols (*Paul et al., 2008*). Brightfield images were acquired at 20 X on a SlideView VS200 slide scanner (Olympus). Coronal sections containing the decussation of the anterior commissure were selected for analysis and quantified using QuPath software. Measurements from three consecutive slices were averaged per mouse.

## Immunostaining

Desired sections were identified by comparison to a mouse brain atlas (*Franklin and Paxinos, 2013*). Three 30 µm coronal sections from the motor cortex, spaced at 120 µm intervals and containing the rostral decussation of the corpus callosum, were used for immunostaining. Mounted sections were boiled in Na-Citrate buffer (10 mM, pH 6.4) for 45 min, blocked for 1 hr in 5% normal goat serum (NGS) in 0.3% PBS-T, incubated overnight at 4 °C with primary antibodies (**Key Resources Table**), then with appropriate AlexaFluor-conjugated secondary antibodies for 1 hr at RT, and counterstained for

10 min with DAPI. Controls lacking primary antibody were used to confirm staining specificity. Fluorescent images were acquired at 10 X on a SlideView VS200 slide scanner (Olympus). Semi-automatic cell counts for DAPI+, NeuN+, and PV +stained cells were obtained using QuPath by an experimenter blinded to genotype.

## Dendritic arbor reconstructions and dendritic spine density analysis

Mice of all genotypes examined were intercrossed with mice bearing a *thy1-GFP* transgene (M line; *Feng et al., 2000*). Fluorescent images of neurons contained within 200 µm coronal sections were acquired at 20 X for arbor reconstructions (Zeiss LSM900) and at 100 x on an UltraVIEW VoX spinning disc confocal microscope for dendritic spines. Dendrites were traced and measured using the Simple Neurite Tracer Fiji plugin (SNT) in ImageJ (NIH). Spines were manually counted from max projection z-stack images on secondary apical and basal dendritic branches of L5 PNs.

## Electron microscopy

Electron microscopy of synapses in M1 layer 5 was performed as previously described (*Katrancha et al., 2019*). Asymmetric excitatory synapses were identified as an electron-dense post-synaptic density (PSD) apposed to a presynaptic terminal containing synaptic vesicles. Measurements of synaptic features, including synapse density, PSD length, spine head area, presynaptic bouton area, and synaptic vesicle distribution, were quantified in ImageJ as previously described (*Montesinos et al., 2015*; *Dong et al., 2018*; *Levy et al., 2018*; *González-Forero et al., 2012*), from 5 to 18 fields of view (~50 µm$^2$) per mouse by a reviewer blinded to genotype.

## Electrophysiological recordings

Acute slices were prepared from mice at P35-42 as previously described (*Katrancha et al., 2019*) with modifications noted here. Coronal slices of M1-M2 cortex were cut at 360 µm in ice-cold N-Methyl-D-glucamine-aCSF (NMDG-aCSF, in mM): 120 NMDG, 2.5 KCl, 7 MgSO$_4$, 1.25 NaH$_2$PO$_4$, 0.5 CaCl$_2$, 28 NaHCO$_3$, 13 glucose, 7 sucrose, saturated with 95% O2-5% O2 at 300–320 mOsmol/L, pH 7.4 on a vibratome (Leica). Slices were recovered in ACSF (in mM): 120 NaCl, 2 KCl, 1.2 NaH2PO4, 26 NaHCO3, 1.3 MgSO4, 1.6 CaCl2, 11 glucose, 2 ascorbic acid, 4 Na-lactic acid, 2 Na-pyruvate saturated with 95% O2-5% O2 at 300–320 mOsmol/L, pH 7.4, at 32 °C for 20 min followed by 1 hr recovery and recordings at RT. Recorded signals were acquired at a 100 kHz sampling rate and low pass filtered at 6 kHz. Cells were excluded from analysis if series resistance changed >20%.

Excitatory postsynaptic currents (EPSCs) were evoked (e) by a tungsten bipolar stimulating electrode placed in M1 L2/3 with stimulus intensities that yielded 40–50% of the maximal response. AMPAR-eEPSC amplitudes were measured at the peak of the EPSC recorded at –70 mV holding potential. NMDAR-eEPSC amplitudes were recorded from the same neurons at +40 mV and measured at 40ms after the stimulus artifact when AMPAR-eEPSCs have decayed.

Miniature EPSCs (mEPSCs) were recorded in whole-cell patch mode with 1 µM tetrodotoxin, 5 µM strychnine, and 20 µM bicuculline methiodide (BMI) in the external solution. AMPAR-mEPSCs were recorded at –70 mV with 50 µM D-2-Amino-5-phosphonovaleric acid (D-AP5) to block NMDAR. NMDAR-mEPSCs were recorded at +45 mV with 5 µM Cyanquixaline (CNQX) to block AMPAR. Gamma-aminobutyric acid receptor and glycine receptor inhibitory postsynaptic currents (GABAR/GlyR-mIPSCs) were recorded at +15 mV with D-AP5 and CNQX. Amplitude and frequency of miniature events were detected using Mini-Analysis software (Synaptosoft, Decatur, GA) and inspected visually.

Paired-pulse stimulation was applied with interstimulus intervals (ISIs) of 35, 60, 100, 200, or 300ms. The amplitude of the second eEPSC (EPSC$_2$), evoked by the second of the paired-pulse stimulations, was divided by eEPSC$_1$ amplitude to obtain a paired-pulse ratio (PPR). Long-term potentiation (LTP) of evoked EPSCs (recorded at –70 mV) was induced by theta-burst stimulation (TBS, five trains of four pulses at 100 Hz in 200ms intervals, repeated four times with an interval of 10 s) in the presence of 5 µM strychnine and 20 µM BMI.

Estimations of the readily releasable pool (RRP) size and synaptic vesicles (SVs) depletion rate were made from recordings of AMPAR-eEPSCs using high-frequency stimulation (HFS, 40 Hz) trains of 15 pulses to deplete the RRP. Trains were analyzed to estimate glutamate probability of release (Pr) and RRP for each neuron using the decay method as previously described (*Ruiz et al., 2011*;

*Thanawala and Regehr, 2016*) using OriginLab 10 software. RRP recovery was approximated as fractional recovery using initial train stimulation with interstimulus intervals of 0.1, 2, 5, and 10 s, followed by a single stimulus. Fractional recovery for each ISI was calculated as (eEPSCs2-eEPSCt1ss)/(eEPSCt1-eEPSCt1ss), where eEPSCs2-amplitude of the stimulus applied after the train, eEPSCt1-first amplitude of the train, eEPSCt1ss-steady state amplitude of the train (last 5 responses). 1 mM kynurenic acid was added to prevent AMPA receptor saturation; 50 µM vD-AP5 was added to block NMDA receptors, 5 µM strychnine and 20 µM BMI to block GABA and glycine receptors.

PPR and train recordings for the rescue experiments were performed as described above with the addition of 100 µM NSC23766 into the recording solution with a 5-min incubation period to allow for efficient slice penetration.

## Quantitative proteomics - sample preparation

Cortices from male P21 mice (four per genotype) were removed and immediately frozen in liquid nitrogen. Mouse brain tissues were transferred to Covaris tissueTUBE TT1 (Part No. 520001, Plug Part No. 520006), kept on dry ice, and flash frozen in liquid nitrogen prior to cryopulverization using a Covaris CP02 cryoPREP Automated Dry Pulverizer (Part No. 500001). The resulting powdered samples were transferred to 1.5 ml microcentrifuge tubes and lysed in 300 µl lysis buffer (8 M urea in 50 mM Tris pH 8.0, 150 mM NaCl, 1 mM EDTA and protease phosphatase inhibitor) on ice. Lysates were sonicated using AFA Covaris E220 (Part No. 500239). The sonicated samples were centrifuged for 30 min at 10,000 rpm. Clear supernatants were transferred to new tubes and quantified with the BCA assay (Thermo). Lysates were reduced with 10 mM TCEP and alkylated with 10 mM iodoacetamide for 30 min in the dark at room temperature. 1 mg lysates were diluted to 4 M urea with 200 mM Tris pH 8.0 and digested with Lys-C (1:50 protease:protein ratio) for 90 min at 37 °C. Samples were then diluted to less than 2 M urea with 200 mM Tris pH 8.0 and digested with trypsin at the same ratio overnight at 37 ° C. The samples were acidified with 1% formic acid and desalted using 50 mg per 1 cc Sep-Pak (Waters). A 250 µg aliquot of each sample was frozen and dried, then stored at –80 ° C until tandem mass tag (TMTpro 16, Thermo lot# WI325918) peptide labeling according to manufacturer instructions.

Desalted 16-plex TMT-labeled peptides (5 mg total) were subjected to basic reverse-phase (RP) chromatography. Peptides were separated using a 4.6 mm x 250 mm RP Zorbax 300 A Extend-C18 column (Agilent, 5 mm bead size) on an Agilent 1260 series HPLC. The 96 min gradient of solvent A (2% ACN, 5 mM ammonium formate) to solvent B (90% ACN, 5 mM ammonium formate) stayed at 0% B for 7 min, from 0% to 16% B over 6 min, from 16% to 40% over 60 min, from 40% to 44% over 4 min, from 44% to 60% over 5 min and maintained at 60% B over 14 min. Flow rate was 1 ml/min. 90 fractions were collected and concatenated down to 24 fractions. 5% of each final fraction was used for proteome analysis. The remaining 95% of each of the 25 fractions was concatenated to 13 fractions. These 13 fractions were IMAC enriched using Ni-NTA Agarose beads as described for phosphoproteome analysis (*Mertins et al., 2018*).

## Quantitative proteomics - data acquisition

Proteome analysis was performed using an EASY-nLC 1200 UHPLC coupled to an Orbitrap Eclipse mass spectrometer (Thermo Fisher Scientific). The online peptide separation was performed using a 25 cm x 75 mm i.d. silica picofrit capillary column (New Objectives) packed with 1.9 mm ReproSil-Pur C18-AQ beads (Dr. Maisch GmbH). The 110 min method, 84 min effective gradient of solvent A (2% ACN, 0.1% FA) and solvent B (80% ACN, 0.1% FA). The gradient started at 2% B and increased to 10% B over 1 min, from 10% to 50% B over 84 min, from 50% to 72% B over 9 min, from 72% to 90% B over 1 min, stayed at 90% B over 5 min, then dropped to 60% B over 1 min and maintained at 60% B for 9 min. The flow rate was at 200 nl/min. The Eclipse mass spectrometer performed data-dependent acquisition in positive ion mode. MS1 spectra scanned a range of 350–2000 m/z at 60000 resolution, with maximum injection time of 50ms and 100% AGC target. MS2 spectra scanned at first mass of 110 m/z at 50,000 resolution. The cycle time for MS2 scan was set for 2 s for charge state of $1 < x < 6$, with isolation window of 0.7 m/z. The AGC target for MS2 was 60% with a maximum injection time of 105ms. An empirically determined normalized collision energy was set to 32. Dynamic exclusion was set to 20 s.

## Quantitative proteomics - data analysis

All raw files were searched using Spectrum Mill (Agilent). MS2 spectra were searched against the Uniprot Mouse Database (20171228, 47,069 mouse entries, 264 common laboratory contaminants) with a mass tolerance of 20 ppm for both the precursor and product ions. The enzyme specificity was set for Trypsin and allowed up to three missed cleavages. The fixed modification was carbamidomethylation at cysteine. TMT labeling was required at lysine, but peptide N-termini were allowed to be either labeled or unlabeled. Allowed variable modifications for whole proteome datasets were acetylation of protein N-termini, oxidized methionine, deamidation of asparagine, and pyroglutamic acid at peptide N-terminal glutamine, with a precursor MH +shift range of $-18$–$64$ Da. The false discovery rate was less than 1%. Protein identifications were discarded if the protein was only observed by a single peptide. Protein subgroups were collapsed to the proteoform with the most evidence.

## Gene set enrichment analysis

Mouse UniProtIDs for identified proteins were converted to human orthologs and ranked by signed $\log_{10}$(nominal p-value) (with sign indicating direction of fold-change from WT). GSEA 4.3.3 (*Subramanian et al., 2005*; *Mootha et al., 2003*) was used to run the GSEA Preranked tests against all gene sets in the Human MSigDB Collection C2 v2023.2. SynGO 1.2 was used to identify synaptic ontologies for differentially expressed proteins with nominal p-value <0.05.

## Data analysis

Statistical analyses were performed using GraphPad Prism 10. Data in bar graphs are presented as mean ± SEM, with individual data points graphed when applicable. Sample size 'n' is annotated within the bars or in the Figure legend for each group. Distributions were tested for normality and outliers (>1.5 x the interquartile distance for behavioral data, or by Prism ROUT method at Q=5% for other metrics) were removed before proceeding with statistical tests. Specific details of statistical tests performed, adjusted with post-hoc Bonferroni test for multiple comparisons (MC) where appropriate, are noted in the Figure legends. Significance was defined by a p-value less than 0.05: $^{ns}$p <0.1; *p<0.05; **p<0.01; ***p<0.001; ****p<0.0001.

## Acknowledgements

We are grateful to Xianyun Ye, Suxia Bai, Andrew Boulton, Chris Kaliszewski, and Xiao-Yuan Li for expert technical support and Bruce Herring, Katherine Roche, Dick Mains and Betty Eipper for formative discussions.

## Additional information

### Funding

| Funder | Grant reference number | Author |
| --- | --- | --- |
| American Heart Association | 20POST35210428 | Yevheniia Ishchenko |
| National Institutes of Health | R56MH122449 | Anthony J Koleske |
| National Institutes of Health | R01MH133562 | Anthony J Koleske |
| National Institutes of Health | R01MH132685 | Anthony J Koleske |
| Simons Foundation | Pilot Award | Anthony J Koleske |

The funders had no role in study design, data collection and interpretation, or the decision to submit the work for publication.

## Author contributions

Yevheniia Ishchenko, Conceptualization, Data curation, Formal analysis, Supervision, Funding acquisition, Validation, Investigation, Visualization, Methodology, Writing – original draft, Project administration, Writing – review and editing; Amanda T Jeng, Data curation, Formal analysis, Validation, Investigation, Visualization, Methodology, Writing – original draft, Writing – review and editing; Shufang Feng, Khanh K Nguyen, Data curation, Formal analysis, Validation, Investigation, Visualization; Timothy Nottoli, Validation, Investigation, Visualization, Methodology; Cindy Manriquez-Rodriguez, Data curation, Formal analysis, Validation, Investigation, Visualization, Methodology; Melissa G Carrizales, Matthew J Vitarelli, Formal analysis, Investigation, Visualization; Ellen E Corcoran, Investigation; Charles A Greer, Conceptualization, Resources, Data curation, Supervision, Validation, Investigation, Visualization, Methodology; Samuel A Myers, Resources, Data curation, Supervision, Funding acquisition, Methodology; Anthony J Koleske, Conceptualization, Resources, Data curation, Supervision, Funding acquisition, Methodology, Project administration, Writing – review and editing

## Author ORCIDs

Yevheniia Ishchenko ⓘ https://orcid.org/0000-0002-8631-9586
Anthony J Koleske ⓘ https://orcid.org/0000-0003-4105-9909

## Ethics

All animal work was performed in compliance with federal guidelines and approved by the Yale Institutional Animal Care and Use Committee (IACUC protocol #2025-07912).

Reviewer #1 (Public review): https://doi.org/10.7554/eLife.103620.3.sa1
Reviewer #2 (Public review): https://doi.org/10.7554/eLife.103620.3.sa2
Author response https://doi.org/10.7554/eLife.103620.3.sa3

# Additional files

## Supplementary files
MDAR checklist

## Data availability
All data generated by this study are included in the manuscript and supporting files. Further information and requests for unique resources and reagents should be directed to and will be fulfilled by Anthony J Koleske (anthony.koleske@yale.edu).

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
