## [Editor Report · eLife Assessment]

This study explores how heterozygosity for specific neurodevelopmental disorder-associated TRIO variants affects brain function in mice. The authors conducted thorough analyses on mouse lines harboring TRIO-variants associated with autism spectrum disorder, schizophrenia, and bipolar disorder, and the results provide **compelling** evidence demonstrating unique alterations of each variant in synaptic functions and behavior. These findings highlight a **fundamental** aspect of TRIO variants contributing to brain functions and neuropsychiatric disorders.

---

## [Referee Report · Reviewer #1 (Public review)]

Summary:

This study explores how heterozygosity for specific neurodevelopmental disorder-associated Trio variants affects mouse behavior, brain structure, and synaptic function, revealing distinct impacts on motor, social, and cognitive behaviors linked to clinical phenotypes. Findings demonstrate that Trio variants yield unique changes in synaptic plasticity and glutamate release, highlighting Trio's critical role in presynaptic function and the importance of examining variant heterozygosity in vivo.

Strengths:

This study generated multiple mouse lines to model each Trio variant, reflecting point mutations observed in human patients with developmental disorders. The authors employed various approaches to evaluate the resulting behavioral, neuronal morphology, synaptic function, and proteomic phenotypes.

---

## [Referee Report · Reviewer #2 (Public review)]

Summary:

The authors generated three mouse lines harboring ASD, Schizophrenia, and Bi-polar-associated variants in the TRIO gene. Anatomical, behavioral, physiological, and biochemical assays were deployed to compare and contrast the impact of these mutations in these animals. In this undertaking the authors sought to identify and characterize the cellular and molecular mechanisms responsible for ASD, Schizophrenia, and Bi-polar disorder development.

Strengths:

The establishment of TRIO dysfunction in the development of ASD, Schizophrenia, and Bi-polar disorder is very recent and of great interest. Disorder-specific variants have been identified in the TRIO gene, and this study is the first to compare and contrast the impact of these variants in vivo in preclinical models. The impact of these mutations was carefully examined using an impressive host of methods. The authors achieved their goal of identifying behavioral, physiological, and molecular alterations that are disorder/variant specific. The impact of this work is extremely high given the growing appreciation of TRIO dysfunction in a large number of brain-related disorders. This work is very interesting in that it begins to identify the unique and subtle ways brain function is altered in ASD, Schizophrenia, and Bi-polar disorder.

Weaknesses:

(1) Most assays were performed in older animals and perhaps only capture alterations that result from homeostatic changes resulting from prodromal pathology that may look very different.

(2) Identification of upregulated (potentially compensating) genes in response to these disorder specific Trio variants is extremely interesting. However, a functional demonstration of compensation is not provided.

(3) There are instances where data is not shown in the manuscript. See "data not shown". All data collected should be provided even if significant differences are not observed.

I consider weaknesses 1 and 2 minor. While they would very interesting to explore, these experiments might be more appropriate for a follow up study. The missing data in 3 should be provided in the supplemental material.

Revised Manuscript:

All of my above concerns were well addressed by the authors in the revised submission.

---

## [Author Response]

The following is the authors’ response to the original reviews

**eLife Assessment**
This study provides useful findings about the effects of heterozygosity for Trio variants linked to neurodevelopmental and psychiatric disorders in mice. However, the strength of the evidence is limited and incomplete mainly because the experimental flow is difficult to follow, raising concerns about the conclusions' robustness. Clearer connections between variables, such as sex, age, behavior, brain regions, and synaptic measures, and more methodological detail on breeding strategies, test timelines, electrophysiology, and analysis, are needed to support their claims.

We appreciate the opportunity to address the constructive feedback provided by eLife and the reviewers. Below, we respond to the overall assessment and individual reviewers' comments, clarifying our experimental approach, addressing concerns, and providing additional details where necessary.

We thank the editors for highlighting the significance of our findings regarding the effects of Trio variant heterozygosity in mice. We acknowledge the feedback concerning the experimental flow and agree that clarity is paramount. To address these concerns:

(1) Connections between variables: The word limit of the initial submission constrained our ability to provide adequate details and connections between variables. We have revised the manuscript to explicitly outline and extend explanations and the relationships between sex, age, behavior, brain regions, and synaptic measures, ensuring that the rationale for each experiment and its relevance to the overall conclusions are improved.

(2) Methodological details: The Methods section of our initial submission was condensed, with key details provided in the Supplemental Methods section. We have merged all into an extended section to improve clarity. We have expanded our description of breeding strategies, test timelines, electrophysiological protocols, and data analysis methods in the revised Methods section. We believe the additions have enhanced the transparency and reproducibility of our study and ensured full support of our conclusions.

(3) Experimental flow: We have revised and extended our results, methods, and discussion sections to clarify the rationale and experimental design to guide readers through the experimental sequence and rationale.

We are confident these revisions address the concerns raised and enhance the robustness and coherence of our findings.

**Public Reviews:**

**Reviewer #1 (Public review):**
Summary:This study explores how heterozygosity for specific neurodevelopmental disorder-associated Trio variants affects mouse behavior, brain structure, and synaptic function, revealing distinct impacts on motor, social, and cognitive behaviors linked to clinical phenotypes. Findings demonstrate that Trio variants yield unique changes in synaptic plasticity and glutamate release, highlighting Trio's critical role in presynaptic function and the importance of examining variant heterozygosity in vivo.Strengths:This study generated multiple mouse lines to model each Trio variant, reflecting point mutations observed in human patients with developmental disorders. The authors employed various approaches to evaluate the resulting behavioral, neuronal morphology, synaptic function, and proteomic phenotypes.Weaknesses:While the authors present extensive results, the flow of experiments is challenging to follow, raising concerns about the strength of the experimental conclusions. Additionally, the connection between sex, age, behavioral data, brain regions, synaptic transmission, and plasticity lacks clarity, making it difficult to understand the rationale behind each experiment. Clearer explanations of the purpose and connections between experiments are recommended. Furthermore, the methodology requires more detail, particularly regarding mouse breeding strategies, timelines for behavioral tests, electrophysiology conditions, and data analysis procedures.

We appreciate the reviewer’s recognition of the novelty and comprehensiveness of our approach, particularly the generation of multiple mouse lines and our efforts to model Trio variant effects in vivo.

Weaknesses

(1) Experimental flow and rationale and connection between variables: We have expanded on the connections between behavioral data, neuronal morphology, synaptic function, and proteomics in the Results and Discussion sections to clarify how each experiment informs the reasoning and the conclusions and to highlight the relationships between sex, age, behavior, and synaptic measures.

(2) Methodological details: Our initial Methods section was formatted to be short to fulfill word limits on the submitted version, with additional details provided in the Supplemental Methods section. We have merged our Methods and Supplemental Methods sections and expanded on our breeding strategies, test timelines, electrophysiological protocols, and data analysis. We believe these additions enhance the transparency and reproducibility of our study.

(3) Recommendations for the authors: We thank Reviewer #1 for providing several recommendations to improve our manuscript. We have addressed their comments in the revision, as detailed below, adding key experiments that bolster our findings.

**Reviewer #2 (Public review):**
Summary:The authors generated three mouse lines harboring ASD, Schizophrenia, and Bipolar-associated variants in the TRIO gene. Anatomical, behavioral, physiological, and biochemical assays were deployed to compare and contrast the impact of these mutations in these animals. In this undertaking, the authors sought to identify and characterize the cellular and molecular mechanisms responsible for ASD, Schizophrenia, and Bipolar disorder development.Strengths:The establishment of TRIO dysfunction in the development of ASD, Schizophrenia, and Bipolar disorder is very recent and of great interest. Disorder-specific variants have been identified in the TRIO gene, and this study is the first to compare and contrast the impact of these variants in vivo in preclinical models. The impact of these mutations was carefully examined using an impressive host of methods. The authors achieved their goal of identifying behavioral, physiological, and molecular alterations that are disorder/variant specific. The impact of this work is extremely high given the growing appreciation of TRIO dysfunction in a large number of brain-related disorders. This work is very interesting in that it begins to identify the unique and subtle ways brain function is altered in ASD, Schizophrenia, and Bipolar disorder.Weaknesses:(1) Most assays were performed in older animals and perhaps only capture alterations that result from homeostatic changes resulting from prodromal pathology that may look very different.(2) Identification of upregulated (potentially compensating) genes in response to these disorder-specific Trio variants is extremely interesting. However, a functional demonstration of compensation is not provided.(3) There are instances where data is not shown in the manuscript. See "data not shown". All data collected should be provided even if significant differences are not observed.I consider weaknesses 1 and 2 minor. While they would be very interesting to explore, these experiments might be more appropriate for a follow-up study. I would recommend that the missing data in 3 should be provided in the supplemental material.

We are grateful for the reviewer’s recognition of our study’s significance and methodological rigor. The acknowledgment of Trio dysfunction as a novel and impactful area of research is deeply appreciated.

Weaknesses:

We agree that focusing on older animals limits insights into early-stage pathophysiology. However, our goal in this study was to examine the functional impacts of *Trio* heterozygosity at an adolescent stage and to reveal the ultimate impact of these alleles on synaptic function. Our choice of age aligns with our objectives. Future studies of earlier developmental stages will be beneficial and complement these findings.

Functional compensation:

We tested functional compensation through rescue experiments in *+/K1431M* brain slices using a Rac1-specific inhibitor, NSC23766, which prevents Rac1 activation by Trio or Tiam1. Our finding that direct Rac1 inhibition normalizes deficient neurotransmitter release in *+/K1431M* mice strongly suggests that increased Rac1 activity drives this phenotype.

Data not shown:

We will incorporate all previously shown data into the Supplemental Materials, even when results are nonsignificant. We agree that this ensures full transparency and facilitates a more comprehensive evaluation of our findings.

**Recommendations for the authors:**

**Reviewer #1 (Recommendations for the authors):**
(1) In Figure 1K-N, the lack of observed differences in +/M2145T mice across all tests raises questions about its validity as a BPD model. Furthermore, the differences in female behavior data compared to males, as shown in the Supplemental section, lack clarification-specifically, whether these variations are due to sex differences or sample size disparities, which is not discussed. Additionally, it's unclear if the same mice were used in tests K through L-N, as the reported numbers differ without explanation; if relevant, any mortality should be reported. Given the observed body weight differences, it is important to display locomotor data, despite the mention of no change in open field results. Lastly, a detailed breeding strategy and timeline for behavioral testing would enhance clarity.

We thank Reviewer 1 for recognizing these confusing points in our behavioral data and seek to add clarification in our Revision as below:

(a) We have revised the text to emphasize our goal to evaluate the impact of NDD-related *Trio* alleles that have discrete and measurable effects on brain development and function, and not to model specific NDDs (e.g. ASD, SCZ, or BPD). The three specific *Trio* mutations were chosen based on strong evidence of these mutations impairing the biochemical functions of Trio. We reasoned our approach would reveal how impairing Trio in different ways – i.e. altering protein level or GEF1/GEF2 function – and under genetic conditions (heterozygosity) that mimic those found in individuals with Trio-related disorders impacts brain development and function. The lack of behavioral phenotypes in *+/M2145T* mice is indeed intriguing, especially given the alterations in electrophysiology and biochemistry experiments. It remains possible that further behavioral analyses of these mice will reveal behavioral phenotypes.

(b) Given that the prevalence and clinical presentation of individuals with various NDDs are influenced by sex, it is possible that the behavioral differences we see in male versus female *Trio* variant mice reflect human sex difference phenotypes. We have reorganized the Figure panels to clarify these sex differences in behaviors (new Fig. 2, Supp. Fig. 2). We focused on the most significant behavioral phenotypes shared by both sexes in the main text, or in males alone, as our anatomical and electrophysiological experiments were restricted to males to reduce variation due to estrus. The observed behavioral sex differences are not likely due to sample size disparities as power analyses were performed for all experimental results to ensure adequate sample size. A comprehensive study of the mechanisms underlying these behavioral findings merits examination but is outside the scope of this study.

(c) All mice were subjected to all behavioral tests described. No sudden mortality was observed during the behavioral experiments. Outliers in post-hoc statistical analyses were removed, which explains the apparent sample size differences between behavioral tests. We have revised the Data analysis section in our Methods to include these details (Lines 216-289, 450-457).

(d) Results of the open field test have been added to the Supplemental Data (new Supp. Fig. 2) and Results (Lines 532-537)

(e) The Methods section was expanded to include more detail on the breeding strategy (Lines 98-106). A timeline for behavioral testing has also been included in the Figures to enhance clarity (new Fig. 2A).

(2) In Figure 2A-E, head width and brain weight showed significant differences, but not body weight, how come the ratio does not change? Comparing with female results in Supplementary Figure 2A-E, it does show a difference between males and females. It is essential to clarify which sex authors use in all follow-up experiments, including synapse, transmission, and plasticity. Since the males and females have different phenotypes, why do the authors focus on males only? The E plot has no data points on the bar graph. In Figure 2I, it lacks example images for all four conditions.

We greatly appreciate this Reviewer’s attention to details in our brain and body weight data and revised the manuscript to address these concerns.

(a) The ratios of head width/body weight were calculated for each individual mouse. Hence the distribution of the ratio data (old Fig. 2D; new Fig. 3D) differs from the distribution of head width or body weight data alone (old Fig. 2A, 2C, resp.; now Fig. 3A, 3C), and therefore can affect the p-value for statistical significance. The body weight of *+/M2145T* males is 21.217 ±0.327 g, while for WT males is 21.745 ±0.224 g, a non-significant decrease of 0.528 g (adjusted p=0.3806). These values have been added to the Fig 3. figure legend (Lines 1020-1034) for clarity.

(b) Similar to the behavioral experiments in comment (1), we observed sex differences in head width, brain weight, and body weight in *Trio* heterozygous variant mice compared to WT counterparts. The differences in the ratios of head width/body weight or brain weight/body weight were the same for both males and females (i.e. head width/body weight ratio is decreased in *+/K1431M* mice compared to WT regardless of sex, and brain weight/body weight ratio is decreased in both *+/K1431M* and *+/K1918X* mice compared to WT regardless of sex). These findings affirm the impact of *Trio* mutations on these phenotypes across both sexes. We have modified the text to draw more attention to this key point (Lines 554-566 and 777-801).

(c) All experiments (excluding behavior and weight data) were performed in males only to minimize the variation in spine and synapse morphology and physiological activity that can occur due to estrus. We have clarified this in the ‘Animal Work’ section of the Methods (Lines 103-106) as well as in the Figure Legends.

(d) We thank the Reviewer for pointing out Fig. 3E lacks individual data points on the bar graph. Fig. 3E has been modified to now include the brain weight/body weight ratio for each individual mouse rather than across the population, to be consistent with the calculation of head width/body weight ratio (see point 2a).

On original submission, only a representative WT image was selected due to space constraints. The figure (new Fig. 3H and 3K) and figure legend have been revised to include representative traces for all genotypes examined.

(3) In lines 315-320, "None of the Trio variant heterozygotes exhibited altered dendritic spine density on M1 L5 pyramidal neurons compared to WT mice on either apical or basal arbors (Supplementary Figure 3L, M). Electron microscopy of cortical area M1 L5 revealed that synapse density was significantly increased in +/K1918X mice compared to WT (Figure 3A, B), possibly due to a net reduction in neuropil resulting from smaller dendritic arbors." The proposed explanation does not adequately address the observed discrepancy between spine density and synapse density reported in these two experiments. A more thorough analysis is needed to reconcile these conflicting findings and clarify how these distinct measurements may relate to each other in the context of the study's conclusions.

We acknowledge the apparent discrepancy between our dendritic spine density data, which is unchanged from WT for all three *Trio* variant heterozygotes, and our synapse density data, which showed an increase in *+/K1918X* M1 L5 compared to WT. We have expanded the explanation for this discrepancy below and added this to the Discussion (Lines 802-811):

a) Because spine density can vary by dendritic branch order and distance from the soma, only protrusions from secondary dendritic arbors of M1 L5 pyramidal neurons were quantified for consistency in analyses. However, all synapses meeting criteria were quantified in EM images, regardless of where they were located along an individual neuron’s arbors. It is possible that the density and distribution of spines along other arbors are different between genotypes but was not captured in our current data.

b) *+/K1918X* L5 pyramidal neurons are smaller and less complex than WT neurons, especially in the basal compartment corresponding to L5 where EM images were obtained, consistent with the smaller brain size and reduced cortical thickness of *+/K1918X* mice. We posit that due to their smaller dendritic field size, L5 neurons pack more densely contributing to the increased synapse density observed in *+/K1918X* M1 L5 cortex. Consistent with this hypothesis, we observed a trend toward increased DAPI+ cell density in M1 L5 of *+/K1918X* neurons (Supp. Fig. 3N).

(4) In Figure 4, one potential rationale for measuring AMPAR mEPSC frequency is to infer synapse density changes. However, the findings show no frequency change in +/K1431M and +/K1918X, with an increase only in +/M2145T, which contradicts Figure 3 results indicating a trend toward increased density across variants.This inconsistency is confusing, especially since the authors claim to follow the methodology from the study "Trio Haploinsufficiency Causes Neurodevelopmental Disease-Associated Deficits"; yet, the observed mEPSC amplitude differs significantly from that study, while the frequency remains unaffected. Additionally, the NMDAR mEPSCs reflect combined AMPAR and NMDAR responses at positive holding potentials, with peak amplitude dominated by AMPAR. This inconsistency between holding potential results is unclear, as frequency should theoretically align across negative and positive potentials. For accurate NMDAR mEPSC measurement, it would be optimal to assess amplitude 50 ms post-initial peak and, if possible, increase the holding potential to enhance the driving force given the typically low signal of NMDAR response.

We thank the Reviewer for highlighting these important points.

a) Previous work from our lab and others demonstrate that Trio regulates synaptic AMPA receptor levels, which is why we chose to focus on AMPAR-mediated evoked and miniature EPSC frequencies and amplitudes in the current study. We acknowledge Reviewer 1’s comment on seemingly contradictory results regarding AMPAR mEPSC frequency and synapse density; however, the unchanged AMPAR mEPSC frequency in *+/K1431M* and *+/K1918X* mice is consistent with our finding of unaltered dendritic spine density in these mice compared to WT (Supp. Fig. 4L,M). The differences between dendritic spine counts and synapse density is addressed in Response (3) above.

b) While synapse density changes can be inferred from AMPAR mEPSC frequency, mEPSCs are also measures of spontaneous neurotransmitter release changes especially in the absence of changes in synaptic numbers. Notably, the increased mEPSC frequency in the *+/M2145T* variant is linked to enhanced spontaneous release, not to spine or synapse density changes. These findings are reinforced by increase in counts of synaptic vesicles, calculated PPR changes, and estimates of the Pr and RRP from HFS train analysis. We have included these points in the Discussion (Lines 861-863).

c) While it is tempting to compare the current study to our previously published conditional Trio haploinsufficiency model, we highlight key distinctions that may underlie phenotypic differences between these two mouse models. First, our prior model used a *NEX-Cre* transgene to ablate one *Trio* allele from excitatory neurons only beginning at embryonic day 11. In contrast, our *Trio* variants are expressed in all cell types throughout development, akin to the genetic variants found in individuals with *TRIO*-related disorders. Second, the *Trio* variant mice in this study are on a C57BL/6 background, while the *Trio* haploinsufficient mice were on a mixed 129Sv/J X C57BL/6 background. These differences in the current study may explain why some measures, such as mEPSC amplitude, may not align with those from the *Trio* conditional haploinsufficiency model.

d) Recordings were performed using specific inhibitors to isolate AMPA and NMDA mEPSCs; these missing methodological details have now been clarified in the updated Methods section (Lines 353-360).

(5) In Supplementary Figure 4, the sample traces indicate a higher NMDA/AMPA ratio, raising the question of whether the AMPA EPSC amplitude changes, as this could reflect PSD length. In Figure 4B, the increased AMPAR mEPSC amplitude in the +/K1918X condition compared to WT suggests an enhanced postsynaptic response, yet the PSD length is reduced in Figure 3C. Can the authors provide a potential hypothesis to explain this?

We appreciate the Reviewer’s feedback. Yes, both evoked and miniature recordings indicate increased AMPAR amplitudes in the *+/K1918X* variants compared to WT. While PSD length is often linked to synaptic strength, the observed reduction in PSD length in EM PSD length reduction in *+/K1918X* synapses is small (~6% of WT) and clearly does not correlate with significant changes in synaptic strength. We also note that the whole cell recordings of mEPSCs represent input from all active synapses on the neuron, while PSD length is measured only in synapses of the L5.

(6) In Figure 4, synaptic plasticity appears to decrease to around 50% of baseline; could this reduction be attributed to LTD, or might it result from changes in pipette resistance? Additionally, is the observed potentiation due to changes in presynaptic release probability? Measuring paired-pulse ratio (PPR) before and after induction would clarify this aspect.

We thank the Reviewer for highlighting these important points.

a) We used a well-established theta burst stimulation method for LTP induction in M1 L5 pyramidal neurons. This protocol reliably evokes LTP in WT neurons, as shown in Fig. 5J and K. Both *+/K1431M* and *+/K1918X* variants exhibit a slight but discernible increase in evoked excitatory postsynaptic currents (eEPSCs), indicative of the initiation of LTP. Although this increase is smaller compared to WT, the presence of potentiation indicates that long-term depression (LTD) is an unlikely explanation for the observed reduction.

b) To rule out the influence of technical artifacts, pipette resistance was carefully monitored before and after LTP induction. Any cells exhibiting resistance changes exceeding 20% during electrophysiological recordings were excluded from the analysis, ensuring that fluctuations in pipette resistance did not confound LTP measurements. These technical details are denoted in the Methods (Lines 344-346 and 364-366).

c) The potentiation in the *+/M2145T* variant may stem from increased release probability (Pr) and greater synaptic vesicle availability, but is beyond the scope of this work. We agree this is an intriguing question, not only for *+/M2145T* but also for *+/K1431M* mice. Future studies should address this, ideally using models where the *Trio* variant is selectively introduced into the presynaptic neuron.

(7) In lines 377-380, "The +/M2145T PPR curve was unusual, with significantly reduced PPF at short ISIs, yet clearly increased PPF at longer ISI (Figure 5A, B) compared to WT." The unusual PPR observed at the 100 ms ISI appears unexpected. Can the authors provide an explanation for this anomaly? This finding could suggest atypical presynaptic dynamics or modulation at this specific interval, which may differ from typical synaptic behavior. Further insights into possible mechanisms or experimental conditions affecting this result would be valuable."The decreased PPF at initial ISI in +/M2145T mice correlated with increased mEPSC frequency (Fig. 4A-C), suggestive of a possible increase in spontaneous glutamate Pr." If this is the case, it raises the question of why the increased PPR at the initial ISI in +/K1431M does not correspond to the result shown in Figure 4C. This discrepancy suggests that factors beyond initial presynaptic release probability might be influencing the observed synaptic response, or that compensatory mechanisms could be affecting PPR and mEPSC frequency differently in this variant. Further clarification on the interplay between these measurements would help resolve this inconsistency.

We appreciate the Reviewer’s critical reading and genuine interest on this phenotype in *+/M2145T* mice.

a) The unusual shift of the PPR in *+/M2145T* at ISI 100ms is fascinating and will require significant additional experimentation that lies beyond the scope of this report to address. We propose it results from altered presynaptic regulators, including increased Syt3 and reduced RhoA activity. Notably, Syt3 influences calcium-dependent SV replenishment, which can cause similar PPR defects (Weingarten DJ et al., 2022); this is now included in the Discussion. (Lines 915-918).

Weingarten DJ, Shrestha A, Juda-Nelson K, Kissiwaa SA, Spruston E, Jackman SL. Fast resupply of synaptic vesicles requires synaptotagmin-3. Nature. 2022 Nov;611(7935):320-325. doi: 10.1038/s41586-022-05337-1. Epub 2022 Oct 19. PMID: 36261524.

b) Thank you for raising the concern in clarity of this statement "The decreased PPF at initial ISI in +/M2145T mice correlated with increased mEPSC frequency (Fig. 4A-C), suggestive of a possible increase in spontaneous glutamate Pr." We have edited the sentence to be more clear (Lines 701-703). First, the K1431M and M2145T variants impact different TRIO catalytic activities disrupting distinct GTPase pathways and differentially affecting presynaptic regulators, which can lead to non-overlapping phenotypes. Also, we expand our discussion that *+/K1431M* variant data suggest increased AMPAR numbers and fewer silent synapses (Lines 850-855), potentially increasing AMPAR mEPSC frequency and masking the expected decrease in spontaneous release (Lines 905-910). Further experiments are needed, ideally using mixed cultures with TRIO variants in presynaptic neurons with synapses on WT neurons, as minimal stimulation variance analysis in slices would be inconclusive due to its reflection of both Pr and silent synapse changes, similar to mEPSC frequency.

(8) In Figure 5, there is no evidence demonstrating that the NSC inhibitor functions specifically in the +/K1431M condition without affecting other conditions. To verify its specificity, the authors should test the NSC inhibitor's effects across other conditions in parallel, including a control group. Additionally, cumulative RRP measurements should be provided for a more comprehensive assessment of the inhibitor's impact on synaptic function.

We appreciate the Reviewer’s feedback.

a) Previous studies have shown that Rac1 activity can bidirectionally regulate synchronous release probability (Pr). We used the Rac1-specific inhibitor NSC23766 (NSC) to test how Rac1 inhibition impacted the neurotransmitter release deficits observed in *+/K1431M* mice. We also added control experiments testing the impact of NSC on WT slices. These new experiments are now presented in new Fig. 8 of the revised manuscript, with expanded details in the Results (Lines 737-750) and Discussion (Lines 892-900).

b) To estimate Pr and the RRP, we employed the Decay method as described by (Ruiz et al., 2011), which does not rely on cumulative EPSC plots for RRP estimation. This approach was chosen to account for the initial facilitation in these synapses and fits are done using EPSCs plotted against stimulus number. Additional details have been provided in the Methods section (Lines 367-373).

Ruiz R, Cano R, Casañas JJ, Gaffield MA, Betz WJ, Tabares L. Active zones and the readily releasable pool of synaptic vesicles at the neuromuscular junction of the mouse. J Neurosci. 2011 Feb 9;31(6):2000-8. doi: 10.1523/JNEUROSCI.4663-10.2011. PMID: 21307238; PMCID: PMC6633039.

(9) Given the relevance to NDD, specifying the age window of the mice used is crucial. It is confusing that the synaptic function studies were conducted at P42, while the proteomic analysis was performed at P21. Could the authors clarify the rationale behind using different age points for these analyses? Consistency in age selection, or an explanation for this variation, would help in interpreting the developmental relevance of the findings.

P42 was chosen as the age as it represents young adulthood, by which time clinical features will have already presented in individuals with neurodevelopmental disorders. Our prior studies of *NEX-Cre Trio-/-* mice found significant measurable differences from WT at this age, after neuronal migration, differentiation, synaptogenesis and pruning have occurred. An earlier developmental timepoint, P21, which coincides with juvenile age in mice, was chosen for proteomics studies to identify earlier changes and potentially targetable and modifiable mechanisms that could influence the phenotypes we observed in older mice. The experiments in P42 versus P21 mice were originally two independent lines of investigation that converged in the current study.